# Bulk and Single-Cell RNA Sequencing Elucidate the Etiology of Severe COVID-19

**DOI:** 10.3390/ijms25063280

**Published:** 2024-03-14

**Authors:** Łukasz Huminiecki

**Affiliations:** Institute of Genetics and Animal Biotechnology, Polish Academy of Sciences, Postępu 36A, Jastrzębiec, 05-552 Magdalenka, Poland; l.huminiecki@igbzpan.pl

**Keywords:** COVID-19, SARS-CoV-2, RNA-seq, scRNA-seq, inflammation, pneumonia, neuronal COVID-19, acute respiratory distress syndrome, multisystem inflammatory syndrome

## Abstract

Coronavirus disease 2019 (COVID-19) is a type of pneumonia caused by the SARS-CoV-2 coronavirus. It can cause acute pulmonary and systemic inflammation, which can lead to death in severely ill patients. This study explores the potential reasons behind severe COVID-19 and its similarities to systemic autoimmune diseases. This study reviewed unbiased high-throughput gene expression datasets, including next-generation and single-cell RNA sequencing. A total of 27 studies and eight meta-analyses were reviewed. The studies indicated that severe COVID-19 is associated with the upregulation of genes involved in pro-inflammatory, interferon, and cytokine/chemokine pathways. Additionally, changes were observed in the proportions of immune cell types in the blood and tissues, along with degenerative alterations in lung epithelial cells. Genomic evidence also supports the association of severe COVID-19 with various inflammatory syndromes, such as neuronal COVID-19, acute respiratory distress syndrome, vascular inflammation, and multisystem inflammatory syndrome. In conclusion, this study suggests that gene expression profiling plays a significant role in elucidating the etiology of severe COVID-19.

## 1. Introduction

In the introduction, I will provide background information on COVID-19, its emergence, and its impact on global health. Next, I will highlight the severity of the disease, its symptoms, and its implications for multiple organ systems. I will also discuss the genomic characteristics of SARS-CoV-2 and its implications for studying COVID-19. Finally, I will emphasize the importance of functional genomics in understanding the etiology of COVID-19.

COVID-19 is a new type of pneumonia caused by SARS coronavirus 2 (SARS-CoV-2). The disease first emerged in Wuhan in 2019 and has, since, spread globally. It was officially described as a new disease in 2020, with its respiratory symptoms, including acute respiratory disease syndrome (ARDS), becoming more defined [1,2,3,4,5]. It was determined that SARS-CoV-2, a coronavirus of probable zoonotic origin, is the infectious agent of COVID-19 [6]. COVID-19 can cause a wide range of symptoms, with approximately 15% of patients becoming so ill they require hospitalization. The fatality rate among hospitalized COVID-19 patients was several times higher than that of influenza [7]. Prolonged COVID-19 [8] is a post-acute form of the disease [9], which can cause inflammation and deterioration in the function of multiple organs, such as the lungs, heart, gastrointestinal tract, neurological system, blood vessels, and reproductive system. In severe cases, patients can develop ARDS, which requires mechanical ventilation. Respiratory failure, sepsis, acute cardiac injury, or heart failure are common causes of death due to ARDS. Uncovering the mysteries behind COVID-19 is no easy feat. However, unbiased genomics datasets can unlock crucial insights that help us better comprehend this enigmatic disease.

The natural progression of severe COVID-19 warrants attention. Upon initial infection, the SARS-CoV-2 virus primarily targets the upper airways, replicating within ciliated epithelial cells and triggering inflammation. This early phase commonly manifests as symptoms such as a sore throat, fever, and coughing. Subsequently, the virus progresses to attack the lungs, leading to pneumonia. This stage is marked by severe pulmonary and systemic inflammation, significantly damaging lung tissue. Alveolar structures experience hardening, inflammation, and impairment in regard to the gas exchange function. The damage includes fibrotic scarring, reduced respiratory surfactant, and increased lung permeability.

The virus predominantly targets type II alveolar cells (AT2 cells) in the lung’s tiny air sacs. Electron microscopy reveals the presence of coronavirus particles within these cells, evidencing their infection and subsequent damage through senescence [10]. However, substantial evidence indicates the virus’s impact on endothelial cells (ECs). In vivo studies have been used to demonstrate the disease and the damage inflicted by the coronavirus on ECs [11]. Furthermore, research results underscore the ability of the coronavirus to infect and harm various EC types, including human umbilical vein ECs (HUVECs) [12], and ECs observed in an aortic ring assay [13]. The damage inflicted on the EC layer in the lungs can lead to coagulopathy, endotheliopathy, and vasculitis, extending the impact throughout the body [14]. Consequently, SARS-CoV-2 adversely affects numerous cell and tissue types across multiple physiological systems [15].

It has been suggested that COVID-19 is related to autoimmune diseases, based on traditional biochemical and molecular markers [16]. Patients with pre-existing inflammatory or autoimmune conditions, such as cancer, obesity, hypertension, cardiovascular disease, type 2 diabetes, or connective tissue disorders, are at a higher risk of fatal outcomes if autoantibodies are present. It is possible that a high viral load in various tissues can cause the body to produce high levels of autoantibodies against multiple host proteins, which then attack cells across many tissues, resulting in systemic inflammation. Indeed, autoantibodies, vascular inflammation, and an inflammatory state of ECs, are typical systemic symptoms of extended COVID-19 [17,18]. These changes occur over several weeks, suggesting transcriptional and epigenetic modifications.

It is worth noting that the genome sequence of SARS-CoV-2 was made publicly available in early 2020 [3]. The genome deposited in GenBank, under RefSeq accession number NC_045512.2, consists of a 29,903 base pair long single-stranded RNA. It contains 11 protein-coding genes, including a multi-unit replicase, a multi-unit protease, and four structural proteins. One of these proteins is protein S, a viral surface glycoprotein that can attach to a mammalian receptor called the angiotensin-converting enzyme 2 (ACE2). ACE2 is a protein that can be found attached to cell membranes in the lungs, small intestine, arteries, or veins, and can sometimes be present in high numbers [19]. Another viral protein, E, is located on the surface of the viral envelope. Protein M is a glycoprotein that is found in the cell membrane. Protein N is a phosphoprotein situated on the nucleocapsid.

On the other hand, additional open reading frames (ORFs), likely non-coding, may have a regulatory function [20]. It was found that the viral RNA had 41 RNA modification sites [21]. Viral transcripts were highly expressed and dominated host gene expression. Surprisingly, the virus’s genome also encoded a furin cleavage site, absent from its closest known phylogenetic relatives. The furin site might enhance the tropism of the virus toward a more significant number of tissue and cell types [22,23]. Therefore, there was enough genomic information about SARS-CoV-2 to investigate COVID-19 using functional genomics as early as 2020.

Here, I review the etiology of COVID-19 using unbiased functional genomics datasets. I had access to many such datasets, available in public databases. For instance, Gene Expression Omnibus has hundreds of genome-wide expression profiles pertinent to COVID-19 (refer to Table 1). Messenger RNAs (mRNAs) were typically isolated from in vitro cell cultures or bulk tissue samples and subjected to next-generation sequencing (NGS). Alternatively, single-cell RNA sequencing (scRNA-seq) was used to investigate specific tissues.

## 2. Functional Genomic Studies of Tissues and Cells Provide Unbiased Evidence to Understand the Etiology of COVID-19

I will start by mentioning 11 simple studies that have used RNA sequencing (RNA-seq) on cell lines or whole tissues to investigate the causes of COVID-19. These studies are listed in Table 2 and focus on examining the inflammatory changes in specific tissues using bulk samples. For example, the studies focused on changes in peripheral blood mononuclear cells—PBMCs [24,25,26,27] during severe COVID-19, within lung cells/airways, pancreatic islets [28], cerebrospinal fluid—CSF [29], or in conjunctival epithelium [30].

Next, I will present 16 more advanced studies using scRNA-seq to investigate specific tissues. You can find these studies listed in Table 3. The studies focused on tissues such as PBMCs [31,32,33,34,35,36,37,38,39], broncho-alveolar lavage fluid—BALF [40,41], the brain [42], lungs, and lung epithelium [43,44,45], heart [43], liver [43], vascular endothelium [45], and kidneys [43].

Finally, Section 3 and Table 4 review eight meta-analyses, five examining ACE2 gene expression in the lungs, airways [46,47,48,49], or kidneys [50], using scRNA-seq datasets. In one example, a meta-analysis created a reference library for immune cells in the average body and inflammatory diseases [51]. Another example, examined the similarities in gene expression between COVID-19 and cancer [52]. I also reviewed a meta-analysis of nine scRNA-seq studies focused on PBMCs [53].

**Table 2 ijms-25-03280-t002:** Examples of bulk-sample RNA-seq studies of COVID-19. The studies related to the transcription of the SARS-CoV-2 virus included details about the biological models and sample types used and the conclusions drawn from the research. In addition, the studies provided specific information such as the PubMed unique identifier (PMID), dataset identification number (ID), impact factor (IF) of the journal, number of citations, and the technology used by the high-throughput platform. On average, the studies were cited 472 times.

Year of Publication and Reference	Specification of the Biological Model and Sample Type	Main Conclusions of the Study	(1) GEO ID. (2) PMID. (3) Dataset Size. (4) Number of Samples.	(1) IF. (2) Citations (According to the Elsevier Abstract and Citation Database (Scopus). Retrieved on 31 July 2023).	Platform
2020 [54].	Human lung cancer cell lines were infected in vitro with four common viruses: H1N1 influenza A virus (IAV), human respiratory syncytial virus (RSV), human parainfluenza virus 3 (HPIV3), and SARS-CoV-2.	Transcriptional responses did not appear to stop the replication of SARS-CoV-2. Instead, they suggested a protracted inflammatory response.	(1) GSE147507. (2) PMID 32416070. (3) 2.7 Mb. (4) 110.	(1) IF = 67. (2) 2771 citations.	Illumina NextSeq 500 (*H. sapiens*).
2020 [24].	Researchers used a gene expression platform to analyze the gene expression of immune cells extracted from patients with active COVID-19 infections.	A study presented a comprehensive atlas of immune modulations during COVID-19 and identified three significant immunotypes related to disease severity.	(1) GSE152418. (2) PMID 32788292. (3) 1.6 Mb. (4) 34.	(1) IF = 48. (2) 629 citations.	Illumina NovaSeq 6000 (*H. sapiens*).
2020 [25].	Researchers conducted a longitudinal study on PBMCs from thirteen COVID-19 patients. There were five time points profiled for each patient. In total, 358,930 cells were analyzed using scRNA-seq (single-cell RNA sequencing) technology, with an average of 10,900 cells per sample.	COVID-19 was associated with dynamic changes in expression patterns within PBMCs. Severe disease induced both hypoxic and pro-inflammatory signaling, along with impaired activation of the interferon pathway.	(1) GSE161777. (2) PMID 33296687 (3) 0.5 Mb. (4) 101.	(1) IF = 43. (2) 189 citations.	Illumina MiSeq and Illumina NovaSeq 6000 (*H. sapiens*).
2020 [26].	PBMCs were collected from seven hospitalized patients with COVID-19, including four with ARDS, as well as six healthy controls.	Infection with SARS-CoV-2 is characterized by strong downregulation of human leukocyte antigen (HLA) of class I and class II, and interferon-driven inflammatory reactions in monocytes.	(1) GSE150728. (2) PMID 32514174. (3) 372 Mb. (4) 13.	(1) IF = 83. (2) 896 citations.	Illumina NovaSeq 6000 (*H. sapiens*).
2021 [55].	Scientists studied lung cells of deceased COVID-19 patients and compared them to normal cells.	Lungs of COVID-19 patients showed inflammation and impaired T-cell response. Diseased lungs had fewer epithelial cells and more macrophages, monocytes, neuronal cells, and fibroblasts.	(1) GSE171524. (2) PMID 33915568. (3) 1601.4 Mb. (4) 27.	(1) IF = 70. (2) 213 citations.	Illumina NovaSeq 6000 (*M. musculus*).
2021 [56].	Blood, lungs, and airways of dead COVID-19 patients were profiled.	Results of the gene expression screening highlighted the importance of changes in immunological gene expression. Not only epithelial cell types, but also infiltrating immune cells, contributed to the signal.	(1) GSE147507. (2) PMID 33782412. (3) 1800.9 Mb. (4) 110.	(1) IF = 5. (2) 75 citations.	Illumina NextSeq 500 (*H. sapiens*).
2021 [29].	CSF was collected from seven men and one woman with neuro-COVID, aged from 53 to 82.	This study identified the expansion of populations of dedifferentiated monocytes and exhausted CD4+ T-cells in the CSF of neuro-COVID patients.	(1) GSE163005. (2) PMID 33382973. (3) 83 Mb. (4) 16.	(1) IF = 43. (2) 84 citations.	Illumina NextSeq 500, or Illumina NovaSeq 6000 (*H. sapiens*).
2021 [28].	Cell culture of human pancreatic islets.	After infecting cultured pancreatic islets with the SARS-CoV-2 virus, it was observed that the number of insulin-secretory granules in β-cells was reduced, leading to impaired glucose-dependent insulin secretion. The infection caused morphological, transcriptional, and functional changes in the β-cells of the islets, resulting in impaired glucose-stimulated insulin secretion.	(1) GSE159717. (2) PMID 33536639. (3) 3.9 Mb. (4) 4.	(1) IF = 21. (2) 318 citations.	HiSeq 4000 instrument (Illumina).
2022 [27].	The study investigated tissue samples from COVID-19 patients. Blood samples were taken for transcriptome profiling from 21 patients. In total, 57,049 single-cell transcriptomes of PBMCs were sequenced.	Endothelial injury and thrombotic events are common in COVID-19 and linked to increased myeloid cell activity.	(1) GSE208337. (2) PMID 35895716. (3) 573.4 Mb. (4) 105.	(1) IF = 13. (2) 7 citations.	Illumina NovaSeq 6000 (*H. sapiens*).
2022 [30].	Organotypic clusters of conjunctival epithelium were isolated from deceased patients’ eyes and cultured. The transcriptomes of 15,821 cells from three infected and three uninfected cultures were processed.	There was no evidence of productive viral replication in ocular epithelial cells; however, there was evidence of the entry of the coronavirus into these cells.	(1) GSE191232. (2) PMID 35750043. (3) 78 Mb. (4) 7.	(1) IF = 7. (2) 2 citations.	

**Table 3 ijms-25-03280-t003:** Examples of scRNA-seq studies of gene expression related to COVID-19. Table 3 displays examples of scRNA-seq studies that examine gene expression about COVID-19, including information on biological models and sample types. Specifically, the following information was given: dataset identification number (ID), impact factor (IF) of a journal, number of citations, and technology used by the high-throughput platform. The average number of citations was 303.

Year of Publication and Reference	Specification of the Biological Model and Sample Type	Main Conclusions of the Study	(1) GEO ID. (2) PMID. (3) Dataset Size. (4) Number of Samples.	(1) IF. (2) Citations (According to the Scopus Database. Retrieved on 31 July 2023).	Platform
2020 [31].	Five COVID-19 patients and three healthy individuals donated their blood to isolate PBMCs.	A single-cell gene expression atlas was created for both COVID-19 and influenza patients. In COVID-19 patients, three signaling pathways were activated: apoptosis, signal transducer and activator of transcription 1 (STAT1), and interferon regulatory factor 3 (IRF3). COVID-19 patients had an increased number of plasma cells.	(1) CNP0001102 (at China’s National Gene Bank Database (db.cngb.org/cnsa)). (2) PMID 32783921. (3) n/a. (4) n/a.	(1) IF = 43. (2) 206 citations.	DNBelab C4 library and DIPSEQ T1 sequencer.
2020 [32].	PBMCs were obtained from five healthy donors and 13 COVID-19 patients, including those with moderate, severe, and convalescent cases.	COVID-19 induces an acute inflammatory response, with a strong induction of the interferon-alpha pathway. There is also evidence of disorganized interferon response and immune exhaustion in severe cases.	(1) HRA000150 (this is an accession identifier (ID) in the Chinese National Genomics Data Center (NGDC)). (2) PMID 32788748. (3) n/a. (4) 64.	(1) IF = 31. (2) 356 citations.	Illumina NovaSeq 6000 (*H. sapiens*), scRNA-seq.
2020 [40].	This study analyzed immune cells in the lungs of COVID-19 patients and healthy controls. Samples were collected from nine COVID-19 patients and three healthy controls and sequenced for analysis.	A new computational method, Viral-Track, was developed to detect infected cells and showed that SARS-CoV-2 has a detrimental effect on the host immune system.	(1) GSE145926. (2) PMID 32479746. (3) 225.1 Mb. (4) 21.	(1) IF = 67. (2) 285 citations.	Beijing Genomics Institute MGISEQ-2000 platform, scRNA-seq.
2020 [33].	PBMCs from 10 patients, either mild, severe, or control.	Subtle changes in the percentages of neutrophil or monocyte subtypes correlate with the severity of COVID-19.	(1) E-MTAB-9221. (2) PMID 32810439. (4) 10.	(1) IF = 67. (2) 496 citations.	10× Chromium droplet-based platform. Illumina NovaSeq 6000 (*H. sapiens*), scRNA-seq.
2020 [40].	A total of 18 COVID-19 patients (eight with mild symptoms and 10 with severe symptoms) had their PBMCs collected between the third and twentieth day after their symptoms were first diagnosed. In total, 48,266 single-cell expression profiles of PBMCs were analyzed, along with 50,783 control PBMCs from publicly available datasets.	During a SARS-CoV-2 infection, significant changes occur in the myeloid compartment of the immune system. In mild cases, there is an increase in inflammatory monocytes. In severe cases, dysfunctional monocytes are increased, and there is a rapid generation of immature neutrophils due to excessive myelopoiesis.	(1) GSE145926. (2) PMID 32479746. (3) 225.1 Mb. (4) 21.	(1) IF = 67. (2) 805 citations.	10× Chromium droplet-based platform. Illumina NovaSeq 6000 (*H. sapiens*), scRNA-seq.
2020 [41].	The study examined nasopharyngeal and bronchial samples from 19 COVID-19 patients who had a moderate or critical disease, alongside five healthy controls. In total, transcriptional profiles of 160,528 cells were analyzed from 36 samples. The study identified 22 different cell types and states within the populations of epithelial cells and immune cells.	In critical cases, there were stronger interactions between epithelial and immune cells, leading to more severe lung injury, respiratory failure, and tissue inflammation.	(1) EGAS00001004481 (in the European Genome-Phenome Archive). (2) PMID 32591762. (4) 36.	(1) IF = 59. (2) 635 citations.	Illumina NovaSeq 6000 (*H. sapiens*), scRNA-seq.
2020 [35].	PBMCs were collected from COVID-19 patients (hospitalized and non-hospitalized), healthy non-exposed subjects, and from subjects both before and after receiving a flu vaccination.	ScRNA-seq analysis of CD4+ T-cells from 40 COVID-19 patients demonstrated that hospitalization caused an increase in the proportion of cytotoxic follicular helper cells and cytotoxic T helper cells, while reducing the population of regulatory T-cells.	(1) GSE152522. (2) PMID 33096020. (3) 518 Mb. (4) 78.	(1) IF = 67. (2) 300 citations.	Illumina NovaSeq6000 (*H. sapiens*), scRNA-seq.
2021 [36].	A total of 68 samples of peripheral blood were obtained, including 52 samples from 13 patients with severe COVID-19 and five samples from five healthy donors. NKs were isolated from these samples through cell sorting, resulting in 80,325 NKs. Single-cell transcriptomes were then generated from these NKs. The UMAP representation categorized the transcriptomes into different types, such as effector I and II, terminally differentiated, transitional, and proliferating.	Severe COVID-19 is associated with early high serum levels of TGF-beta. High levels of this cytokine early in the course of infection are associated with the inhibition of interferon-driven activation of NKs, and impaired immunological response.	(1) GSE184329. (2) PMID 34695836. (3) 174.8 Mb. (4) 13.	(1) IF = 65. (2) 104 citations.	Illumina NextSeq500 (*H. sapiens*), scRNA-seq.
2021 [43].	Single-cell atlases of 24 lung, 19 heart, 16 liver, and 16 kidney tissue samples from COVID-19 autopsies were generated.	COVID-19 is characterized by failed tissue regeneration and pathological remodeling of diseased tissues. In COVID-19 samples of the lungs, there were more fibroblasts and fewer epithelial cells. The RNA of SARS-CoV-2 was found to be enriched in phagocytic and ECs. In the heart, a reduction in the number of cardiomyocytes and pericytes was observed, with an increase in the number of vascular ECs.	(1) GSE163530. (2) PMID 33915569. (3) 40.4 Mb. (4) 1194.	(1) IF = 65. (2) 320 citations.	NextSeq 550, Illumina NovaSeq 6000, NextSeq 550, Nanostring GeoMx 2020 Broad COVID Platform, scRNA-seq.
2021 [42].	Brain and choroid plexus samples isolated from dead patients with severe COVID-19 were profiled at a single-cell level. A total of 65,309 high-quality nuclei were analyzed.	There is significant evidence to suggest that both COVID-19 and neurodegenerative diseases involve long-term inflammation.	(1) GSE159812. (2) PMID 34153974. (3) 944 Mb. (4) 30.	(1) IF = 70. (2) 270 citations.	Illumina NovaSeq 6000 (*M. musculus*), scRNA-seq.
2021 [37].	The study used scRNA-seq to profile COVID-19, MIS-C, healthy pediatric, and adult individuals. The scRNA-seq results were correlated with disease severity, flow cytometry, antigen receptor repertoire analysis, and serum proteomics.	The research has identified gene expression signatures that are linked to MIS-C. These signatures could be useful in diagnosing inflammatory complications related to COVID-19. The study found that MIS-C tissues showed higher levels of S100A family alarmins, which is a sign of reduced antigen presentation, and increased cytotoxicity in NK and CD8+ T-cells.	(1) GSE166489. (2) PMID 33891889. (3) 826.1 Mb. (4) 54.	(1) IF = 43. (2) 118 citations.	Illumina NovaSeq 6000 (*H. sapiens*), scRNA-seq.
2021 [38].	PBMCs were isolated from a total of 15 donors, including five healthy individuals, seven who had recovered from moderate COVID-19, and three who had recovered from severe COVID-19. The 97,315 resulting PBMC epigenomes were categorized into different subtypes such as monocytes, effector, memory, naive, plasma cells, or NKs using UMAP representation.	This study documented the global chromatin accessibility landscape remodeling in COVID-19 recovered patients. The remodeling patterns indicated the development of immunity through immunological memory against SARS-CoV-2.	(1) HRA000562 (this is an accession ID in the Chinese National Genomics Data Center (NGDC)), PRJNA718009 (this is an accession ID in the Sequence Read Archive (SRA)). (2) PMID 34108657. (3) n/a. (4) n/a.	(1) IF = 28. (2) 43 citations.	Libraries (these are libraries for single-cell T-cell-receptor (TCR) sequencing (scTCR-seq), and TCR-FACS-index-ATAC sequencing (Ti-ATAC-seq)). were processed on HiSeq X Ten platform, Illumina.
2021 [57].	To investigate the role of neutrophils in COVID-19 and determine the effects of dexamethasone.	A COVID-19 single-cell atlas of neutrophil states and the molecular mechanisms of dexamethasone action were created. For more information, visit http://biernaskielab.ca/COVID_neutrophil.	(1) GSE157789. (2) PMID 34782790. (3) 360 Mb. (4) 31.	(1) IF = 53. (2) 90 citations.	Sequencing was performed using Illumina NovaSeq S2 and SP 100.
2022 [39].	PBMCs available from seven children diagnosed with MIS-C, as well as six healthy control samples.	Researchers have found that long non-coding RNAs (lncRNAs) play a crucial role in regulating immune responses to SARS-CoV-2. One example is lncRNA PIRAT, which forms a negative feedback loop with the PU.1 transcription factor. This loop helps promote transcription of alarmins, which are proteins that trigger inflammation in response to the virus. However, COVID-19 promotes inflammation by downregulating PIRAT and upregulating lung cancer-associated transcript 1 (LUCAT1), which is another lncRNA that promotes inflammation.	(1) GSE142503. (2) PMID 35998224. (3) 16.7 Mb. (4) 15.	(1) IF = 10. (2) 6 citations.	Illumina NovaSeq 6000 (*H. sapiens*), scRNA-seq.
2022 [58].	During a period of 2 to 3 months, a total of 209 patients with COVID-19 (in addition to 100 patients with post-acute COVID-19) and 457 healthy individuals were studied. The patients were examined at the time of their initial diagnosis, during the acute phase of the disease, as well as during the recovery phase.	The diagnosis of PASC has been found to be associated with certain risk factors, such as diabetes, viremia, and autoimmune conditions.	(1) E-MTAB-10129. (2) PMID 35216672. (3) 5.2 Mb. (4) 309.	(1) IF = 67. (2) 802 citations.	Illumina NovaSeq 6000.
2023 [45].	The research study conducted single-nucleus RNA sequencing on frozen lung samples from seven patients who died from COVID-19, six pairs of lungs from patients with idiopathic pulmonary fibrosis (IPF), and 12 individuals from a control group. The study identified 38,794 cell nuclei from a vascular fraction, which were further divided into 14 subtypes of ECs. Additionally, there were 38,794 non-vascular nuclei. The primary focus of the study was on ECs and a comparison between IPF and COVID-19.	Genes involved in cellular stress were enriched, alongside a signature of reduced immunomodulation and impaired vessel integrity. There was also specific receptor–ligand interactions that were either enriched or depleted in COVID-19 or IPF.	(1) GSE159585. (2) PMID 35998078. (3) 488 Mb. (4) 53.	(1) IF = 13. (2) 7 citations.	Illumina HiSeq 4000/NovaSeq 6000 (*H. sapiens*), scRNA-seq

**Table 4 ijms-25-03280-t004:** Meta-analyses. This table presents a list of high-impact meta-analyses that utilized RNA-seq and scRNA-seq datasets to study the etiology of COVID-19. The average number of citations per study was 287.

Year of Publication and Reference	Goal	Conclusions	Datasets	Computational Methods	(1) IF. (2) Citations. (3) PMID.
2020 [46].	A comprehensive meta-analysis of scRNA-seq datasets to identify cell subsets expressing ACE2 and, therefore, targeted by SARS-CoV-2.	ACE2 and TMPRSS2 promote cellular entry of SARS-CoV-2. Type I interferons, and to a lesser extent type II interferons, upregulate ACE2. Cells vulnerable to infection were identified in the lungs.	GSE148829, GSE135069, GSE19190, GSE22147.	Re-analysis with Drop-Seq Computational Protocol v2.0, Seurat. Meta-analysis with UMAP, PCA, gene set enrichment, etc.	(1) IF = 65. (2) 1574 citations. (3) PMID 32413319.
2020 [47].	To investigate the impact of smoking on COVID-19 utilizing scRNA-seq gene expression data from lung and airway epithelial samples from humans, mice, or rats.	For ACE2, the levels of protein and mRNA were highly correlated (r = 0.82, *p*-value < 0.0001) across 53 cell lines. Smokers had higher levels of gene expression of ACE2 in the lungs. ACE2 expression was uncorrelated with age or sex. However, inflammation in the lungs was linked to the induction of expression of ACE2. ACE2 was also stimulated in expression by interferon signaling.	GSE132040, GSE53960, GSE53960, GSE34378, GSE53960, GSE44555, GSE132040, GSE6591, GSE80680, GSE1643, GSE18344, GSE13933, GSE22047, GSE64614, GSE76925, GSE79209, GSE121611, GSE122960, GSE134174, GSE75715, GSE39059, GSE135188, GSE57148, GSE103174, GSE2052, GSE47460, GSE43696, GSE16538, GSE3100, GSE11056, GSE86623, GSE41789, GSE32138, GSE32138, GSE47963, GSE100504, GSE51392, and GSE19392, plus some samples from TCGA, the GTEx portal, the Human Cell Atlas, the Human Protein Atlas, and the Single-Cell Expression Atlas.	The meta-analysis was performed using Python, Excel, and GraphPad Prism. Regressions were performed using Python, using ordinary least squares and the statsmodels package. Analysis of single-cell expression data was performed using Python, Scanpy, and Multicore-TSNE packages. Variable genes were prioritized utilizing the Seurat function in Scanpy. The variable genes were then analyzed using PCA.	(1) IF = 14. (2) 275 citations. (3) PMID 32425701.
2020 [48].	To determine expression patterns for ACE2 or other receptors for SARS-CoV-2 in the respiratory mucosa using scRNA-seq data.	The levels of mRNA and protein of ACE2 are very low in the upper airways and in the lungs. However, there is a mechanism dynamically regulating ACE2 expression in response to SARS-CoV-2.	GSE19190, GSE11906, GSE4302, GSE67472, GSE37147, GSE108134, GSE135893, the FANTOM5 dataset, and a few proteomics datasets, including the Human Proteome Map.	The Cell Ranger pipeline, UMAP, Zenbu genome browser, R packages: pheatmap, Seurat, ggplot2.	(1) IF = 25. (2) 107 citations. (3) PMID 32675206.
2020 [50].	In this study, scRNA-seq data were used to investigate how kidney diseases or medications may alter ACE2 expression in kidneys.	ACE2 expression in proximal tubular epithelial cells of the kidney facilitated infection with SARS-CoV-2.	Nephrocell data were archived at http://nephrocell.miktmc.org. COVID-19 kidney data were archived at https://hb.flatironinstitute.org/covid-kidney.	The scRNA-seq data were analyzed according to protocols of the Kidney Precision Medicine Project (see https://www.kpmp.org/for-researchers#protocols).	(1) IF = 8. (2) 52 citations. (3) PMID 33038424.
2021 [51].	In this study, scRNA-seq data were used to identify cellular phenotypes shared across disparate inflammatory diseases.	Similarities in gene expression and signaling between COVID-19 and other inflammatory diseases are uncovered.	GSE134809, GSE122960, GSE145926, GSE155249, GSE47189, GSE147507, GSE168710, phs001457.v1.p1, phs001529.v1.p1, phs001457.v1.p1, SCP259.	A meta-analysis and integration pipeline was constructed, which models and removes the effects of the technology, the tissue of origin, and the donor. The meta-analysis built a reference library for immune cells in a normal body and in different diseases.	(1) IF = 12. (2) 83 citations. (3) PMID 33879239.
2021 [49].	Muus et al. performed a meta-analysis of receptor genes for SARS-CoV-2, by looking at gene expression in a meta-analysis of 31 lung scRNA-seq studies.	An atlas of cell type-specific associations of age, sex, and smoking with expression levels of *ACE2* (and other co-receptors for SARS-CoV-2).	SCP865, SCP895, SCP891, SCP903, SCP871, SCP870, SCP874, SCP878, SCP887, SCP899, SCP898, SCP902, SCP894, SCP869, SCP872, SCP866, SCP1240, SCP1241, SCP868, SCP877, SCP867, SCP897, SCP886, SCP879, SCP860, SCP881, SCP875, SCP876, SCP900, SCP892, SCP890, SCP889, SCP880, SCP882, SCP882, GSE102592, GSE103918, GSE104600, GSE107747, GSE108571, GSE109037, GSE110973, GSE111014, GSE111360, GSE112570, GSE112845, GSE113036, GSE114530, GSE114724, GSE114802, GSE115149, GSE115189, GSE117211, GSE117403, GSE117824, GSE118127, GSE119212, GSE119506, GSE119507, GSE119561, GSE119594, GSE120446, GSE121267, GSE121600, GSE122342, GSE122703, GSE122960, GSE123926, GSE124263, GSE124334, GSE124472, GSE124494, GSE124898, GSE125680, GSE127472, GSE128066, GSE128169, GSE128518, GSE128889, GSE129845, GSE130073, GSE130117, GSE130151, GSE130238, GSE130318, GSE130430, GSE130888, GSE131685, GSE132802, GSE133704, GSE134809, GSE135618, GSE135929, GSE136103, GSE136314, GSE136394, GSE139249, GSE139324, GSE98201.	Data generated by Chromium instrument 10× were integrated using the Cell Ranger pipeline. Datasets were post-processed using Python harmony-pytorch (for batch correction and Leiden clustering) in Scanpy.	(1) IF = 87. (2) 183 citations. (3) PMID 33654293.
2021 [52].	Chen et al. used cell-line and bulk-tissue RNA-seq to detect overlaps between the host transcriptional responses observed in cancer and those observed in COVID-19 in response to SARS-CoV-2.	There were many similarities in host–disease interactions between cancer and COVID-19. In particular, immune infiltration and inflammation were characteristic of both the diseases.	GSE147507, GSE148729, GSE36969, GSE59185, GSE68820, GSE119856, GSE115770, GSE147507, GSE146507, GSE156063.	All data were public and derived either from GEO or from TCGA. Pathway analysis was performed using GO, wikiPathways, SigTerms software. Standard statistical methods were used, for example two-sided *p*-values, log-transformed gene expression values, false discovery rate correction for multiple testing, and heat maps.	(1) IF = 4.3. (2) 12 citations. (3) PMID 33510359.
2021 [53].	Garg et al. set up a multi-dataset that included samples from nine scRNA-seq studies.	The multi-dataset consisted of 159 samples, deriving from seven medical procedures focused on PBMCs and two focused on BALF. Eight out of 20 evaluated hypotheses were confirmed.	PRJCA002413, PRJCA002564, GSE149689, PRJCA002579, GSE150728, GSE145926, GSE147143, and EGAS00001004481.	The Scanpy protocol was used for the meta-analysis. The following steps were included: normalization of the datasets, log transformation of the data, selection of genes variable in expression levels, and PCA. Harmony was used to integrate data from different samples. UMAP was then used to cluster, visualize, and annotate gene expression data by cell type.	(1) IF = 4.3. (2) 12 citations. (3) PMID 34675242.

### 2.1. RNA Sequencing Can Be Performed on Cell Lines or Whole Tissues to Investigate the Underlying Causes of COVID-19

This section serves as a proof of principle. I will demonstrate the potential of RNA sequencing technology in investigating the origin of COVID-19 using two examples.

The first proof of principle was selected for its simplicity as an in vitro study. Blanco-Melo et al. researched the transcriptomes of immortalized human cancer cell lines infected with SARS-CoV-2 and other respiratory viruses, including an influenza strain. Notably, the cancer cell lines used in the study were isolated from lung tumors [54]. The cells in the body reacted to the SARS-CoV-2 virus by activating an inflammatory gene expression program. This means that the virus triggers the production of proteins that cause inflammation. These studies have shown that the levels of chemokines, proteins that attract immune cells to the site of infection, are relatively high in people infected with SARS-CoV-2. However, the levels of interferon type I and III, proteins that help fight viral infections, are relatively low. Furthermore, the studies suggest that the inadequate production of antiviral countermeasures, such as interferons, inflammatory cytokines, and chemokines, also play a role in the development of COVID-19.

However, in vitro studies lack evidence regarding the context of a viral infection in an organism. For instance, we could not determine the immune system’s status or the interactions between the virus and hundreds of cell types in various body tissues.

The second proof of principle was selected because it is a straightforward in vivo RNA sequencing study. Daamen et al. [56] focused on the in vivo activities of immune cells in samples from hospitalized COVID-19 patients. To be more precise, the authors conducted bulk RNA-seq on clinical samples of PBMCs from blood. Daamen et al. also processed postmortem lung tissue and postmortem samples from diseased airways to generate comparable COVID-19 datasets. They analyzed these datasets together and discussed the significance of the expression of immunological genes. Indeed, their findings showed that many genes associated with innate immune responses were increased in expression in infected tissues. Specifically, the mRNA levels of the type I interferon and other genes essential for antiviral immunity were elevated. On the other hand, the gene expression signatures of the adaptive immune response tended to be decreased.

Interestingly, many cell types, such as various epithelia and infiltrating immune cells, were the sources of transcriptional signals related to COVID-19. For instance, Daamen et al. found populations of myeloid-like cells with high inflammatory transcriptional signatures. They also observed insufficient activated natural killer (NK) cells in diseased samples. This deficiency could prevent the clearing of virus-laden cells. Additionally, it was observed that there were not enough regulatory cluster of differentiation in eight positive (CD8+) T-cells in tissues from COVID-19 patients. (CD8+ T-cells usually aid in mediating adaptive immunity.) With the involvement of multiple cell types, it is natural to inquire about the specific cells implicated in inflammatory reactions and those affected by prolonged inflammation.

It is worth noting that several other bulk-sample studies, chosen for their high impact and focus on examining inflammatory changes, are listed in Table 2.

### 2.2. Examples of RNA-seq Studies of Individual Cells in COVID-19

Single-cell analysis using scRNA-seq provides the advantage of identifying changes in cell proportions and differentially expressed genes (DEGs) on a gene-by-gene and cell-by-cell basis. It also allows the description of gene expression heterogeneity across many cells. Infected cells are usually mixed with healthy ones in a tissue for viral infections. Since each infected tissue has many different cell types (such as ECs, stromal cells, infiltrating immune cells, or epithelial cells), single-cell analysis can help identify and understand the behavior of infected cells. In COVID-19 research, many influential and highly cited papers have leveraged scRNA-seq for expression profiling.

#### 2.2.1. Expression Profiling of Single Cells in the Immune System

Studying the immune system is crucial for understanding how our bodies defend against diseases and infections. By examining the expression profiles of individual cells within the immune system, we can gain valuable insights into how these cells function and interact with one another. Individual immune system cells, whether circulating in the blood or present in body fluids, underwent multiple analyses to confirm whether and how COVID-19 contributed to systemic inflammation. Refer to studies listed in Table 3, selected for their high impact, which will be briefly introduced paragraph by paragraph below.

For example, in a study by Chua et al. [41], scRNA-seq was performed on nasopharyngeal and bronchial samples from 19 COVID-19 patients, categorized as either moderately or critically ill. The study also included five healthy controls. The researchers identified significant epithelial cell types, including basal, secretory, ciliated cells, FOXN4+ cells, and ionocytes, along with a subpopulation of epithelial cells exhibiting a robust interferon-gamma response. Additionally, the study revealed 13 different cell types or states of immune cells, such as macrophages, dendritic cells, mast cells, neutrophils, B-cells, T-cells, or NK cells.

The findings indicated that in critical cases of COVID-19, there were more pronounced inflammatory interactions between immune and epithelial cells, resulting in respiratory tissue damage and correlating with the severity of COVID-19. These inflammatory interactions were identified through the expression profiling of ligand–receptor pairs in both epithelial and immune cells using the CellPhoneDB database.

Compared to moderate cases, critical cases exhibited an increased expression of ACE2, an entry receptor for the SARS-CoV-2 virus, in epithelial cells, correlating with interferon signaling among immune cells. Moreover, critical cases demonstrated inflammatory expression profiles of ligand–receptor pairs, with inflammatory macrophages expressing numerous potent chemokines, chemokine ligands, interleukin-8 (IL8), interleukin-1 beta (IL1B), and tumor necrosis factors (TNFs). These factors increased lung injury, respiratory failure, and inflammatory tissue damage.

Zhu et al. [31] conducted a study on the gene expression of PBMCs in patients with COVID-19 and influenza. They observed that in COVID-19 patients, three signaling pathways were activated: the apoptosis pathway, the signal transducer and activator of the transcription 1 (STAT1) pathway, and the interferon regulatory factor 3 (IRF3) pathway. Additionally, T-cells in COVID-19 patients exhibited three distinct mechanisms of active apoptosis: the X-linked inhibitor of apoptosis (XIAP)-associated factor 1 (XAF1) pathway, the TNF pathway, and the Fas receptor pathway. In contrast, influenza patients showed activation of STAT3 and nuclear factor kappa-light-chain enhancers of activated B-cells (NF-kappa-beta) pathways instead of STAT1/IRF3.

The study additionally found an elevation in fully differentiated PBMCs, responsible for producing specific antibodies and B-cells in COVID-19 patients. However, there was a concurrent decrease in lymphocytes due to apoptosis. The upregulated differentially expressed genes (DEGs) in COVID-19 patients encoded pro-inflammatory cytokines, cytokine receptors, or interferon-responsive transcription factors, signifying a pro-inflammatory response within immune cells during COVID-19. In summary, the study suggested that the transcriptional response in immune cells during COVID-19 was characterized by both a reduced magnitude and a pro-inflammatory nature.

Please note that a study by Wilk et al. [26] performed a similar experiment. They sequenced the transcriptomes of PBMCs from seven COVID-19 hospitalized patients, four of whom had ARDS, and six healthy individuals. The study analyzed 44,721 cells, averaging 3194 cells per sample. Uniform manifold approximation and projection (UMAP) identified 30 cell clusters, and the DEGs were calculated for each cluster. The study identified significant differences in the cell populations of monocytes, T-cells, and natural killer (NK) cells between COVID-19 patients and healthy controls.

Additionally, COVID-19 patients exhibited changes in cell proportions, with several cell types experiencing depletion, including a subtype of T-cells, conventional dendritic cells (DCs), plasmacytoid dendritic cells (pDCs), CD16+ monocytes, and natural killer (NK) cells. Patients with ARDS demonstrated significant depletion in DCs, CD16+ monocytes, and NK cells. The study also identified an increase in plasmablasts and the developing neutrophils population, with the highest levels observed in patients with ARDS.

Another group of authors utilized scRNA-seq of PBMCs to investigate immune cells in the blood in more detail. They sequenced and assembled 57,669 high-quality transcriptomes for seven COVID-19 patients and compared them against five healthy controls [24]. UMAP was used to identify up to 25 immune cell subsets in the resulting NGS data. Several subsets of PBMCs were predominantly identified in patients with COVID-19, such as monocytes and T-cells with a high expression of interferon-stimulated genes, including interferon-alpha inducible protein 27 (IFI27), interferon-induced transmembrane protein 3 (IFITM3), or interferon-stimulated gene 15 (ISG15). Additionally, DEGs were identified between similar subsets in the COVID-19 group versus the control group. Antiviral pathways were induced in the COVID-19 group, particularly in monocytes and dendritic cells, as revealed by these DEGs. Interferon expression was closely monitored, and only modest levels of interferon-gamma expression in T and NK cells were detected. Thus, a comprehensive atlas of transcriptional immune modulations in PBMCs in COVID-19 was presented. Lastly, it was confirmed that very high levels of inflammatory cytokines in COVID-19 could lead to severe autoimmune diseases, resulting in shock, multiple organ failure, or respiratory failure.

Bernardes et al. studied immune responses in PBMCs, using a longitudinal approach to explore disease trajectories over time [25]. They analyzed scRNA-seq data from 358,930 cells, with an average of 10,900 cells per sample, including up to four longitudinal samples per patient. The study aimed to identify dynamic changes in gene expression and correlate them with disease severity in COVID-19. Extensive bioinformatics analyses of whole-blood scRNA-seq data were performed at five different time points to achieve this. The analysis identified DEGs, functional trends, networks of transcription factors, and co-expression modules. The observed set of DEGs suggested transcription factor activity related to inflammation and a signature of interferon signaling in COVID-19. However, the interferon response appeared to be defective in critical cases.

In severe cases of COVID-19, a correlation was observed with low oxygen levels, leading to increased red blood cell production and alterations in gene regulation. Survivors of COVID-19 exhibited higher levels of genes associated with platelet production, whereas non-survivors showed elevated levels of pro-inflammatory cytokines. Individuals with severe COVID-19 also displayed an increased presence of plasmablasts, erythroid cells, and interferon-activated megakaryocytes. Pathway analysis using the Reactome database revealed that non-survivors were more prone to developing megakaryocytes, producing platelets, and increasing pro-inflammatory cytokines induced by TRAF6. This suggests a potential link between these molecular pathways and the severity of outcomes in COVID-19 cases.

In another recent study, cytotoxic T-cells were compared between two groups of COVID-19 patients: those with severe illness and those with moderate symptoms [33]. In the initial phase of the investigation, RNA libraries were constructed to represent individual cytotoxic T-cells. Subsequently, single-cell sequencing was conducted, leading to the identification of the M protein of SARS-CoV-2 as a frequent target of the cytotoxic T-cell receptor (TCR). Specifically, the M198–206 amino acid sequence was recognized as an essential epitope, with 81.1% of the libraries (30 out of 37) responding to this epitope. The study also unveiled differentially expressed genes (DEGs) between populations of mature cytotoxic T-cells and naive ones. Notably, serine proteases granzyme A and B (GZMA and GZMB) were identified as crucial for inducing apoptosis in cells under attack by cytotoxicity. The study concluded that T-cells following severe COVID-19 were less adept at a cytotoxic response than those in moderate cases. However, in contrast, another study by Meckiff et al. [35] found a higher abundance of cytotoxic T-cells among PBMCs in hospitalized patients with COVID-19.

In another example, NGS sequencing was conducted on single-cell transcriptomes of PBMCs from ten moderate, six critical, and five fatal cases of SARS-CoV-2 [27]. Fifty-seven thousand and forty-nine single-cell transcriptomes of PBMCs were sequenced, and the authors identified eight functional clusters of cells among the PBMCs. These clusters encompassed a cluster of differentiation four positive (CD4+) T-cells, a cluster of differentiation eight positive (CD8+) T-cells, B-cells, plasma cells, natural killer cells (NKs), conventional dendritic cells, canonical monocytes, and non-canonical monocytes. The study revealed a higher number of transcriptomes derived from myeloid cells in the PBMCs of fatal COVID-19 cases than moderate cases. These myeloid cells exhibited a biased upregulation of a platelet-activating signature. The authors also concluded that endothelial injury and pathological thrombotic events were common in COVID-19, and these events were positively correlated in regard to frequency with the activity of myeloid cells. A related study of PBMCs also suggested that subtle changes in the proportions of myeloid cells correlated with the severity of COVID-19 [33].

In a study conducted by Sinha et al. [57], researchers investigated the impact of dexamethasone on neutrophils in severely ill COVID-19 patients. Dexamethasone, a widely accepted treatment for COVID-19, was administered at a daily dose of 6 mg, either orally or intravenously. The researchers analyzed 15,000 individual cells, comprising an equal proportion of leukocytes and lymphocytes. The study found that dexamethasone enhanced the immunosuppressive qualities of neutrophils, transforming them from information receivers to information providers.

Bost et al. [40] conducted a study on immune cells in the respiratory system, utilizing scRNA-seq on cells isolated from BALF. The patients had either mild or severe COVID-19. The authors developed a computational method called Viral-Track to interpret their datasets, enabling the detection of viral RNAs in large scRNA-seq datasets. The process was benchmarked on next-generation sequencing (NGS) datasets from virus-infected tissues. Using Viral-Track, the researchers compared immune cells from severe cases with those from mild instances of COVID-19. The study found that severe cases of COVID-19 had a profound impact on the immune system, with marked differences in the proportions of different immune cell subtypes, such as myeloid, lymphoid, or epithelial cells. In mild cases, there was an increase in alveolar macrophages and plasmacytoid dendritic cells (pDCs) in BALF isolated from bronchia. In severe cases, tissue-resident alveolar macrophages were replaced with recruited macrophages, monocytes, or neutrophils.

Research conducted by Heming et al. [29] investigated immune cells from the CSF of patients who experienced varying neurological symptoms during COVID-19. These symptoms included headaches, dizziness, cognitive abnormalities, seizures, encephalitis, stroke, or brain hemorrhage. Patients with neuro-COVID had a higher number of dedifferentiated monocytes and exhausted CD4+ T-cells. These exhausted T-cells resulted from chronic overstimulation in a site of active inflammation, leading to reduced effector functions and expression of co-inhibitory receptors. Additionally, there was a decreased interferon response in neuro-COVID compared to viral encephalitis.

Another recent study focused on young individuals diagnosed with multisystem inflammatory syndrome (MIS-C) following COVID-19 [37]. MIS-C is a severe inflammatory condition affecting various internal and external body parts, such as the lungs, heart, brain, eyes, kidneys, or gastrointestinal tract tissues, causing extensive inflammation. Symptoms of MIS-C include fever or a cytokine storm. The study aimed to identify markers that could help diagnose and predict the severity of MIS-C through gene expression analysis.

The researchers utilized scRNA-seq to identify 30 subpopulations of immune cells in PBMCs. Differentially expressed genes (DEGs) were identified, and the subpopulations of PBMCs between MIS-C patients and healthy controls were compared. The study found that MIS-C patients had a higher expression of S100A family alarmins and decreased gene expression signatures characteristic of antigen presentation. Additionally, the study identified an elevated expression level of cytotoxicity genes in natural killer (NK) and CD8+ T-cells.

Finally, a study by Su et al. [58] explored the causes of post-acute sequelae of COVID-19 (PASC) through scRNA-seq and various multi-omics technologies. This research, focusing on PBMCs and featuring an extended follow-up period, identified four risk factors associated with an increased likelihood of developing PASC. These factors include: (1) type 2 diabetes, (2) high SARS-CoV-2 load in the plasma, (3) Epstein–Barr viremia, and (4) the presence of specific autoantibodies.

#### 2.2.2. Expression Profiling of the Lung Tissue

Researchers investigated samples from diseased lungs to gain insights into the impact of COVID-19 on pulmonary health. This exploration aimed to understand the underlying causes of pulmonary inflammation and ARDS, which can be life threatening in severe COVID-19 cases. For instance, a study by Melms et al. [55] utilized scRNA-seq to examine lung samples from COVID-19 patients. The findings revealed highly inflamed lungs, characterized by numerous activated macrophages and reduced T-cell responses. The impaired T-cell activities observed in the study could contribute to fatal outcomes. Additionally, myeloid cells were identified as a significant source of dysregulated inflammation, exhibiting a higher prevalence in diseased lungs than in healthy ones.

Furthermore, it has been observed that COVID-19 can lead to substantial lung fibrosis, and the degree of fibrosis is directly correlated with the duration of the disease. Abnormal tissue regions were found to harbor numerous pathological fibroblasts. Additionally, inflamed lungs exhibited a reduced presence of alveolar type I (AT1) and alveolar type II (AT2) cells, hindering the regenerative capacity of the lung.

A recent study uncovered widespread lung inflammation in COVID-19 patients, impacting multiple organs. Delorey et al. [43] conducted a comprehensive survey involving the generation of a sizeable single-cell RNA atlas from the lungs, kidneys, livers, and hearts of deceased COVID-19 patients. Transcriptome analysis revealed extensive inflammatory damage in diseased lungs, impairing regeneration. Notably, this damage was observed in cells from patients with and without viral RNAs. The impairment of epithelial progenitors hindered the regeneration of alveolar type II (AT2) cells. Moreover, the gene pathways associated with increased transcriptional output in COVID-19 encompassed apoptosis linked with oxidative stress in pericytes, various immune pathways, cell adhesion pathways, and the signaling pathway for fibroblast differentiation.

#### 2.2.3. Expression Profiling of the Brain, Ocular Epithelia, and the Vasculature

In 2021, scRNA-seq was utilized to analyze brain tissues infected with SARS-CoV-2. The study compared cells from the brains of eight COVID-19 patients with cells obtained from 14 healthy individuals. These patients had severe pneumonia with extreme inflammation and had been using mechanical ventilation for over two weeks. The study found that microglia and astrocyte subpopulations in the brain resembled those found in neurodegenerative diseases. Furthermore, genetic susceptibility to COVID-19 was linked to genetic susceptibility to neurological degeneration, including genes associated with impaired cognition, schizophrenia, or depression. Finally, the study concluded that long-term inflammation is a part of brain-linked COVID-19, with inflammation from the lung tissue being transmitted to the brain [42].

Jackson et al. [30] conducted scRNA-seq to analyze an in vitro model of human ocular epithelia infected with the COVID-19 virus. The researchers obtained ocular epithelial cells from the eyes of three female COVID-19 patients, aged 52, 78, and 80, who had donated their eyes for research purposes. The cells were cultured in vitro on mitotically inactivated 3T3 feeder cells. The study results revealed that the COVID-19 virus could be detected in the ocular epithelial cells and that the conjunctival epithelium was susceptible to the SARS-CoV-2 virion. However, the research showed no productive virus replication within the epithelial cells. This means the SARS-CoV-2 virion could infect more cell types than those it could replicate in. Overall, the study suggests that even eyes could be affected by COVID-19.

Scientists have also studied the vascular system during COVID-19 and have found that ECs contribute significantly to COVID-19 inflammation and the invasion of immune cells, vascular leakage, thrombosis, and hypoxia [59]. Recently, de Rooij et al. [45] studied mRNAs from ECs in cases of lethal COVID-19. In the first step of the survey, single-cell transcriptomes from more than 175,000 cell nuclei were sequenced. UMAP analysis of the resulting expression profiles allowed the differentiation of the nuclei into four major classes: epithelial, stromal, endothelial, or from the immune system. A more detailed UMAP analysis identified as many as 14 different endothelial subtypes among 35,000 EC nuclei in the following step. For example, there were arterial, pulmonary, large vessel, systemic vein, proliferating, capillary, or lymphatic ECs. The authors showed that pulmonary ECs of deceased COVID-19 patients were enriched in genes involved in cellular stress. Moreover, there were gene expression signatures in ECs that were suggestive of failed immunomodulation and impaired vessel integrity. The study found more capillary and venous ECs in COVID-19 patients than in controls.

## 3. A Review of Selected COVID-19 Meta-Analyses

Numerous gene expression datasets are available for COVID-19, allowing them to be merged in a meta-analysis (refer to Table 4). This is an exciting prospect since a comprehensive meta-analysis can include re-analysis, recalculation of significance tests, statistical tests of replicability, and testing the robustness of conclusions. This could strengthen the findings about the mechanism of COVID-19.

In their study, Ziegler et al. aimed to identify lung cells expressing ACE2 and transmembrane serine protease 2 (TMPRSS2), two receptors significant for the SARS-CoV-2 virus [46]. The authors first identified cellular targets of SARS-CoV-2 in human and non-human primate lungs, the gastrointestinal tract, and upper airways. ACE2 was found to be the primary target. Interferons were found to increase ACE2 expression. A related meta-analysis investigated ACE2 levels in COVID-19-associated kidney disease [50].

A meta-analysis by Garg et al. [53] re-evaluated 20 hypotheses on the immune response to COVID-19 by integrating nine PBMC datasets [26,32,41,53,60,61,62]. This meta-analysis focused on the characteristics of a single cellular compartment, the immune cells, in COVID-19. The study identified five significant populations of cells, namely lymphoid cells, myeloid cells, B-cells, epithelial cells, and platelets. The study observed that T-cells tended to decrease in number with increasing severity of COVID-19. Furthermore, the study found that interferon type 1 responses tended to increase in COVID-19, and B-cells tended to expand in selected clones in response to antigens from SARS-CoV-2.

Chen et al. [52] compared the transcriptional response in human cell lines infected with SARS-CoV-2 to that observed in cancer samples processed by the Cancer Genome Atlas (TCGA). The virus infected two cell lines: the lung adenocarcinoma cell line (A549) and normal human bronchial epithelial cells (Calu-3). It was found that there are many similarities in gene expression, indicating the presence of similar inflammatory programs.

It is worth noting that smoking dramatically increases the risk of severe COVID-19. In a meta-analysis by Smith et al. [44], lung and airway epithelial samples were examined to explore the impact of smoking on COVID-19. Another aim of this meta-analysis was to identify the cell and tissue types with high levels of ACE2, the receptor for SARS-CoV-2.

In their study, Muus et al. [49] analyzed publicly available scRNA-seq datasets to identify cell type-specific associations of age, sex, or smoking, with the expression levels of ACE2 and other SARS-CoV-2 receptor molecules. To this end, publicly available scRNA-seq datasets were downloaded from the Gene Expression Omnibus (GEO) using the following criteria: (1) unnormalized count data was provided, (2) data was generated with 10× Genomics Chromium platform, and (3) human samples were analyzed. The researchers conducted a comprehensive analysis of 31 scRNA-seq studies on the lungs, involving more than 1.3 million cells from 377 airway and lung samples, collected from 228 individuals. The findings confirmed that the unique expression patterns of cells are involved in the development of COVID-19.

Finally, Zhang et al. [50] identified phenotypes shared across many inflammatory diseases, including COVID-19. By performing a meta-analysis and using an integration pipeline, gene expression and signaling similarities between COVID-19 and other inflammatory diseases were discovered. The effects of technology, the tissue of origin, and the donor were also modeled. A meta-analysis was conducted to build a reference library for more than 300,000 cells in the average body, different inflammatory diseases, and COVID-19. This cross-disease study identified that interferon-gamma and TNF-alpha signaling in macrophages were crucial in the inflammatory phenotype observed in severe COVID-19. The study also found that CXCL10+ CCL2+ inflammatory macrophages were abundant in severe COVID-19.

## 4. The Etiology of Severe COVID-19 in the Light of Gene Expression Data

After reviewing 27 datasets and eight meta-analyses, we have gained a mature understanding of the causes of severe COVID-19. It is important to note that I focused on high-impact RNA sequencing studies on the etiology of COVID-19, which had, on average, 372 citations. These functional genomics studies were well interpreted and published in peer-reviewed journals with high impact. The advantage of expression profiling is that changes in gene expression are likely to be longer term and stable, mediated through epigenetic reprogramming, and are likely to better characterize the etiology of COVID-19 than biochemical markers. Furthermore, long-term COVID-19 can be complicated by several inflammatory syndromes, which gene expression datasets from different tissues and organs, including neuro-COVID, ARDS, or MIS-C, have described.

In general, the lists of differentially expressed genes and gene sets strongly suggest that the development of severe COVID-19 is due to inflammatory mechanisms (refer to Table 5). In Table 6, six pathways are differentially expressed. Frequently, there was an increase in the interferon response, signaling, or transcription factors. Additionally, there was a noticeable inclination towards heightened immune or inflammatory reactions. The expression of cytokines, chemokines, or their receptors, also showed an inclination toward an increase. Furthermore, there was an increase in the signaling of the interleukin-1 family and hypoxic signaling. A decrease in the regulation of the immune system, angiogenesis, and vessel integrity accompanied these increases.

In conclusion, the studies reviewed herein focused on DEGs, host–virus interactions, and the cellular composition of affected tissues and organs. For instance, by using scRNA-seq (Table 7), scientists could identify changes in the proportion of immune or epithelial cells. This provided insight as to why severe COVID-19 has become a dangerous immune disease that affects multiple organs and tissue types. There were frequent changes in the proportions of immune cells, particularly among PBMCs, infiltrating immune cells, or NKs [24,29,31,40,56,64].

According to recent research, severe COVID-19 patients show an increase in the proportions of B-cells (plasmablasts and plasma cells), neutrophils, cytotoxic T-cells, or activated macrophages, while regulatory immune cells, such as regulatory T-cells, dendritic cells, or antigen-presenting cells decrease in proportion. These observations suggest that a robust immune response at the beginning of COVID-19 may lead to a dysregulated immune system that attacks the lungs, causing scarring and fibrosis. This process occurs over 7–10 days. A diagram illustrating this process is presented in Figure 1. More details are provided in Table 7.

Moreover, evidence from transcription indicates the presence of endothelial injury and pathological thrombotic events [27,45].

It is worth noting that SARS-CoV-2 virus transcription can also be verified using RNA-seq and specialized bioinformatics pipelines. This can be conducted in various types of cells present in the bronchi or lungs of COVID-19 samples, such as epithelial cells, lymphocytes, macrophages, or neutrophils. For more details, see Table 8.

**Table 8 ijms-25-03280-t008:** The transcription of SARS-CoV-2 was confirmed in several studies.

Tissue Type	Positive Cell Types	Type of Evidence	Computational Method	Reference
If detected, transcription of the viral genome in swabs or brush specimens was used to confirm the diagnosis of COVID-19.	n/a	The entire SARS-CoV-2 genome sequence was annotated as one viral ‘gene’. The viral gene was appended to the hg19 annotation gtf file. All reads aligning to the SARS-CoV-2 genome per sample were aggregated and divided by the total number of reads in that sample.	Transcripts were aligned to a customized reference genome in which the SARS-CoV-2 genome (RefSeq ID: NC_045512) was added as an additional chromosome to the human reference genome hg19. Viral load was calculated on the raw data matrices output by Cell Ranger.	[41].
BALF from severe and mild COVID-19 patients.	Epithelial cells and macrophages.	Viral mRNAs were identified among scRNA-seq reads that did not map to the human genome.	Viral-Track was an R-based pipeline that utilized the STAR algorithm to align reads to the SARS-CoV-2 genome.	[40].
BALF from two severe COVID-19 patients [62].	Epithelial cells, lymphocytes, macrophages, neutrophils.	Yeskit integrates host gene expression profiles with virus detection.	R and Python-based packages utilizing the STAR algorithm. Yeskit is an R package for data integration, clustering, identification of DEGs, functional annotation, and visualization.	[66].

**Figure 1 ijms-25-03280-f001:**
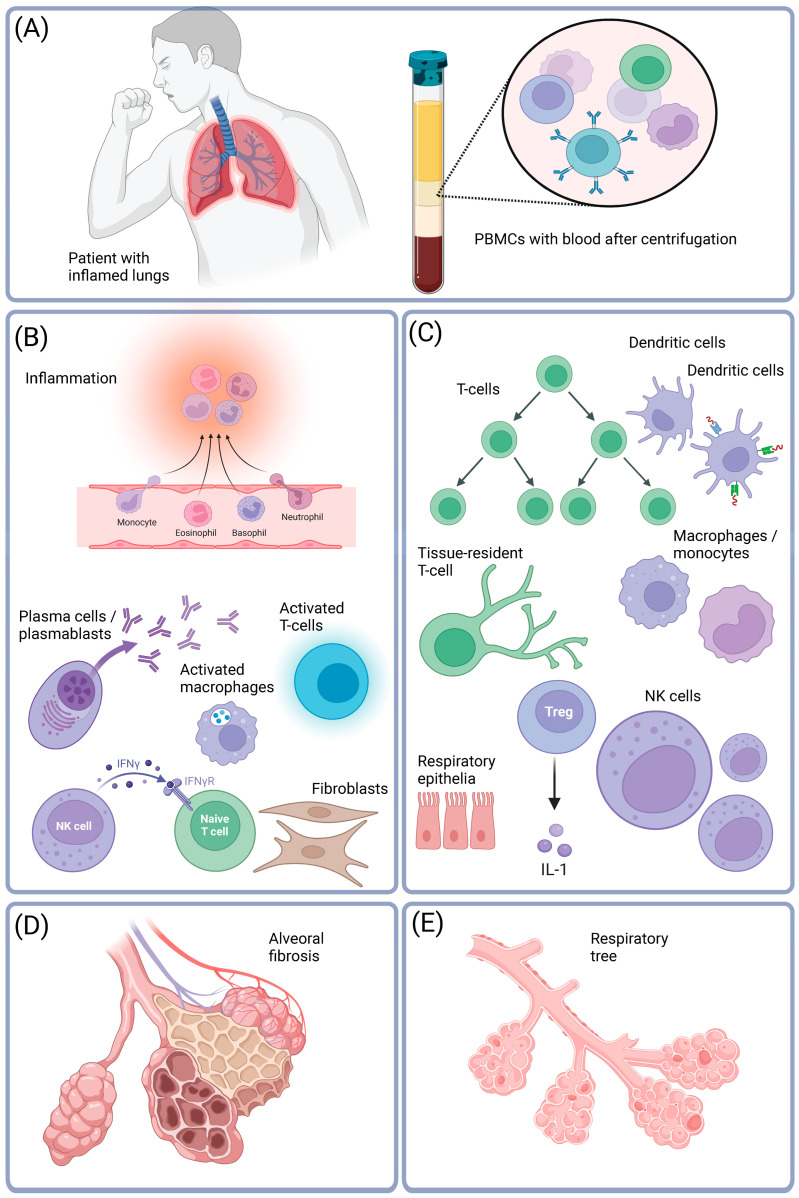
COVID-19 has been associated with significant alterations in immune cell levels, leading to inflammation. The SARS-CoV-2 virus, responsible for COVID-19, primarily infects respiratory cells, and the ensuing immune response can affect various organs and tissues. (**A**) Viral replication in the lungs and the collection of PBMCs. The SARS-CoV-2 virus infects various cell types in the lungs, including epithelial cells, macrophages, lymphocytes, and neutrophils. This viral replication can lead to a cascade of events, triggering inflammation in different organs and tissues. Researchers collect PBMCs from blood samples to gain insights into the immune response. Moreover, scRNA-seq is employed to analyze the gene expression patterns at the single-cell level. (**B**) Immune cell changes in COVID-19. Individuals with COVID-19 show an increase in specific immune cell types, resembling patterns seen in autoimmune diseases and immune aging. Increased cell types include neutrophils, recruited monocytes, plasma cells, plasmablasts, and activated natural killer (NK) cells, which are all associated with inflammation. (**C**) Decrease in regulatory immune cells. Regulatory immune cells, such as regulatory T-cells and naive NK cells, tend to decrease in number during severe COVID-19. Regulatory cells play a crucial role in controlling and balancing the immune response. (**D**) Lung alveoli changes. Severe COVID-19 infections can adversely affect lung alveoli, causing them to harden, become inflamed, and form scars. (**E**) Normal lungs. Alveoli are tiny air sacs responsible for gas exchange in the lungs. Understanding these cellular and molecular changes is pivotal to deciphering the mechanisms underlying severe COVID-19 and developing targeted therapeutic interventions. It highlights the intricate interplay between the virus and the immune system, shedding light on potential avenues for treatment and management strategies. BioRender Figure.

## 5. Methods

Note that analyses listed here generally used standard computational methods such as the STAR RNA-seq aligner [67], the gapped-read aligner Bowtie 2 [68], or the FastQC quality control pipeline. Additionally, popular tools for the analysis of single-cell RNA-seq include Seurat [69], Scanpy [70], and UMAP [69,71,72]. Moreover, DESeq2 was utilized for the identification of DEGs, alongside resources for their annotation, such as the GO database [73], KEGG pathways [74], the STRING database [75], and the CellPhoneDB database [76]. General packages for biostatistics and scientific computing encompass Bioconductor [77], Python [78], NumPy [78], SciPy [79], and the R Project for Statistical Computing. Noteworthy databases comprise the Single-Cell Expression Atlas [80], GEO [81], and FANTOM5 [82]. See also Appendix A for the description of typical computational workflows.

## 6. Conclusions

Severe cases of COVID-19 can last for a considerable period, approximately 17 to 19 days from the time of diagnosis to the time of death [83]. The extended duration is attributed to the prolonged inflammatory response triggered by the SARS-CoV-2 virus. The prolonged inflammatory response can induce significant changes in gene expression patterns, particularly in PBMCs. Moreover, various tissues throughout the body can undergo inflammatory transformations, including changes in cell composition, indicating the systemic impact of severe COVID-19. Understanding the prolonged nature of severe COVID-19 cases and its impact on various aspects of the immune response, gene expression and tissue inflammation are crucial for comprehending the disease’s complexity and the implications for patient health.

## Figures and Tables

**Table 1 ijms-25-03280-t001:** There are many high-throughput datasets related to COVID-19 in the GEO database. If someone searches for SARS-CoV-2 in GEO, they can find up to 800 datasets. These datasets are classified here, based on the species and experimental platform. The total number of datasets available and the number of datasets for *H. sapiens*, *C. sabaeus*, *M. musculus*, and SARS-CoV-2 are provided.

Keyword	Species	Total	NGS	Microarray	ChIP-seq
SARS-CoV-2	All	808	642	22	52
	*H. sapiens*	596	472	16	42
	*C. sabaeus*	14	10	n/a	1
	*M. musculus*	16	14	n/a	2
	SARS-CoV-2	29	20	n/a	n/a
COVID-19	All	536	407	22	33
	*H. sapiens*	536	407	22	33
	*C. sabaeus*	6	5	n/a	1
	*M. musculus*	10	9	1	1
	SARS-CoV-2	19	14	n/a	n/a

**Table 5 ijms-25-03280-t005:** Differentially expressed genes in single-cell RNA-seq datasets in severe COVID-19.

Cell or Tissue Type	Differentially Expressed Genes	Differentially Expressed Pathways or Gene Sets	Reference
PBMCs	Interferon-stimulated gene 15 *(ISG15)* ↑ Interferon-induced protein 44-like *(IFI44L)* ↑ MX dynamin-like GTPase 1 *(MX1)* ↑ XIAP-associated factor 1 *(XAF1)* ↑	Pro-inflammatory cytokines ↑ Cytokine receptors ↑ Interferon-responsive TFs ↑ Response to type 1 interferon signaling ↑ Defense response to virus signaling ↑ Endoplasm and protein unfolding ↑ Regulation of chromosome organization ↑ DNA conformation change ↑	[31].
	Thrombospondin 1 *(THBS1)* ↑	Neutrophil degranulation ↑ Plate activation, signaling, and aggregation ↑ Semaphorin interactions ↑ Crosslinking of collagen fibrils ↑ Interleuking-1 family signaling ↑ Platelet degranulation ↑	[27].
	lncRNA *LUCAT1* ↑ *CXCL2* ↑ *IL-6* ↑ lncRNA *PIRAT* ↓	Hematopoietic cell lineage ↑ Cytokine–cytokine receptor interaction ↑ Chemokine signaling pathway ↑	[39].
	Immunoglobulin heavy constant alpha 1 *(IGHA1)* ↑ Immunoglobulin heavy constant gamma 1 *(IGHG1)* ↑ Lactotransferrin *(LTF)* ↑ Interferon regulatory factor 1 *(IRF1)* ↑ Signal transducer and activator of transcription 3 *(STAT3)* ↑ Hypoxia inducible factor 1 subunit alpha *(HIF1A)* ↑ RAR-related orphan receptor C *(RORC)* ↓	IL-1β and vasodilatory signaling ↑ IFN-related transcripts ↑ Myeloid cell-mediated immunity ↑ Neutrophil degranulation ↑ Erythroid cell differentiation ↑ Cell differentiation pathway ↑ Hypoxic signaling ↑ Inflammation and IFN signaling ↑ Ribosomal structural proteins ↓	[25].
	HLA-DPB1 and HLA-DMA in monocytes ↓	Type I interferon-driven inflammatory signature in monocytes ↑	[26].
	Interferon-induced transmembrane protein 2 (IFITM2) ↑ Interferon-induced transmembrane protein 3 (IFITM3) ↑ Interferon-induced protein 20 (ISG20) ↑ Interferon-induced protein 15 (ISG15) ↑	Interferon gamma response ↑	[41].
	Interferon-ɑ response upregulation ↑	HLA-class II downregulation in CD14+ monocytes ↓	[53].
ECs	Heat shock protein 90 alpha family class A member 1 *(HSP90AA1)* ↑ Heat shock protein family A (Hsp70) member 1A *(HSPA1A)* ↑ TIMP metallopeptidase inhibitor 1 *(TIMP1)* ↑ Fibrillin 1 *(FBN1)* ↑ Matrix metallopeptidase 16 *(MMP16)* ↑ Collagen type XV alpha 1 chain *(COL15A1)* ↑ Indoleamine 2,3-dioxygenase 1 *(IDO1)* ↑ Intercellular adhesion molecule 1 *(ICAM1)* ↓ Interferon regulatory factor 1 *(IRF1)* ↓ Cadherin 5 *(CDH5)* ↓ Integrin subunit beta 1 *(ITGB1)* ↓ Member of RAS oncogene family *(RAP1B)* ↓ Cell division cycle 42 *(CDC42)* ↓ Occludin *(OCLN)* ↓ Vinculin *(VCL)* ↓ Sphingosine-1-phosphate receptor 1 *(S1PR1)* ↓ Protein C receptor *(PROCR)* ↓ Thrombomodulin *(THBD)* ↓	Genes involved in cellular stress ↑ Heat shock proteins ↑ Genes involved in antigen presentation ↑ Hypoxia signaling ↑ Extracellular matrix (ECM) interactions ↑ ECM production/remodeling ↑ Immune system regulation ↓ Vessel maintenance/integrity ↓ Inflammation ↓ Angiogenesis ↓ Cell–cell adhesion ↓ Chemokines/cytokines ↓ TNF and JAK/STAT signaling ↓	[45].
Lungs	ACE2 ↑	Interferon response ↑	[46,47,48].
	ACE2 ↑ TMPRSS2 ↑ CTSL ↑	Interferon response ↑	[49].
Lung cancer cell lines	Interferon gamma receptor 1 (IFNGR1) ↑ Interferon gamma receptor 2 (IFNGR2) ↑ Colony stimulating factor 2 (CSF2) ↑ Colony stimulating factor 3 (CSF3) ↑ C-X-C motif chemokine ligand 1 (CXCL1) ↑ C-X-C motif chemokine ligand 2 (CXCL2) ↑ Interleukin 1 alpha (IL1A) ↑ Interleukin 1 beta (IL1B) ↑ Interleukin 6 (IL6) ↑ Tumor necrosis factor superfamily member 14 (TNFSF14) ↑	Type II interferon signaling ↑ Immune response ↑ Response to stress ↑ Cytokine-mediated signaling ↑ Inflammatory response ↑ Cytokine activity ↑ Extracellular space ↑ Growth factor receptor binding ↑ Response to virus ↑	[52].
Kidney	ACE2 ↑	Interferon response ↑	[50].

↑ signifies upregulation, while ↓ signifies downregulation.

**Table 6 ijms-25-03280-t006:** Effects on gene sets in severe COVID-19.

Differentially Expressed Gene Sets	References
Increased interferon response, interferon signaling, interferon-responsive TFs.	[25,26,31,46,47,48,49,50,52,63].
Increased immune or inflammatory responses.	[26,31,52].
Increased expression of cytokines, chemokines, or receptors.	[31,39,45,52].
Increased interleukin-1 family signaling.	[27,52].
Increased hypoxic signaling.	[25,45].
Diminished immune system regulation, angiogenesis, and vessel integrity.	[45,53].

**Table 7 ijms-25-03280-t007:** Changes in proportions of cell types detected by scRNA-seq in the case of inflammatory COVID-19.

Tissue Type	Reference	Cell Type Increasing in Frequency in Severe COVID-19	Less Common Cell Type
PBMCs	[53].	Plasma cells. Increased B-cell clonal expansion.	Regulatory T-cells.
	[31].	Plasmablasts.	Lymphocytes.
	[63].	Plasmablasts.	n/a
	[27].	Myeloid cells.	n/a
	[26].	Developing neutrophils. CD14+ monocytes. Plasmablasts.	CD16+ monocytes. Plasmacytoid dendritic cells, conventional dendritic cells, and NK cells.
	[35].	Cytotoxic follicular helper cells, and cytotoxic T helper cells.	Regulatory T-cells.
	[65].	Highly cytotoxic NK cells containing high levels of cytotoxic proteins, such as perforin.	Unarmed NK cells.
	[41].	Activated macrophages.	n/a
BALF	[40].	Recruited macrophages, monocytes, or neutrophils.	Alveolar macrophages.
	[41].	Neutrophils.	Basal epithelial cells.
CSF	[29].	Dedifferentiated monocytes, and CD4+ T-cells.	n/a
Lungs	[55].	Fibroblasts, myeloid, and neuronal cells.	Antigen presenting cells, epithelia.

## Data Availability

No new data were created or analyzed in this study. Data sharing is not applicable to this article.

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
