# Peer review of "Bulk and Single-Cell RNA Sequencing Elucidate the Etiology of Severe COVID-19"

_ijms, 2024, doi:10.3390/ijms25063280_

Round 1

Reviewer 1 Report

Comments and Suggestions for Authors

This review provides a comprehensive curation of (single cell? - see below as I am not sure) transcriptomic datasets related to COVID.

Such a curation of datasets could be quite useful for researchers interested in doing meta-analyses of COVID transcriptomics.

___

The title of the paper starts "Single-cell transcriptomics..."

However, throughout the paper a number of transcriptomic datasets are discussed without clarifying whether they are bulk or single-cell data sets. Make sure that each time you mention a single-cell transcriptomic dataset, you specify that it is single cell. Do not use the naked word "transcriptomic" to describe scRNA datasets. If you have any non single-cell transcriptomic datasets, then change you title of your paper to make sure you encompass these datasets.

___

Although the paper category is listed as "Review", it is sufficiently in the ballpark of a "Systematic Review" that the PRISMA guidelines should be followed and a PRISMA checklist included as Supplemental Material.

https://www.equator-network.org/reporting-guidelines/prisma/

As part of following the guidelines, along with the other guidelines, a Methods section should be included, which will focus on inclusion/exclusion guidelines for including a transcript dataset in this review.

___

Style could use some work.

For example, the Abstract currently reads

"Abstract: COVID-19 was a novel inflammatory pneumonia caused by a respiratory infection with a coronavirus named SARS-CoV-2. Severely ill COVID-19 patients could die of acute pulmonary and systemic inflammation. Etiology was unclear, but there was a theory that severe COVID -19 shared many similarities with systemic autoimmune diseases. There was also a theory that the SARS-CoV2 virus could infect many cell- and tissue-types distributed throughout the body. I review the above theories using unbiased high-throughput datasets of gene expression. Studies reviewed used the technology of next generation sequencing, either in bulk or as single -cell RNAs sequencing; ingenious methods for signal deconvolution were then used to identify individual cell types. Moreover, 27 datasets reviewed, together with 8 meta-analyses, suggested that severe COVID-19 induced expression of genes associated with pro-inflammatory signaling, interferon response, as well as cytokine or chemokine signaling. Concurrently, there was frequently an inflammatory change in proportions of cell-types among immune cells in the blood or infiltrating tissues, or degenerative change among epithelia of the lungs. There was also genomic evidence for several inflammatory syndromes associated with COVID-19, such as neuronal COVID, acute respiratory disease syndrome, vascular inflammation, or multisystem inflammatory syndrome. I conclude that etiology of severe covid-19 is now well explained using functional genomics."

When ask Perplexity.AI

Please suggest style improvements to this Abstract for a scientific paper: "<paste in abstract here>", I get

The abstract can be improved by restructuring the sentences to enhance clarity and flow. Additionally, the use of technical terms and jargon should be minimized to improve accessibility. Here's a revised version:

"Abstract: COVID-19, caused by the SARS-CoV-2 coronavirus, is a novel inflammatory pneumonia that can lead to acute pulmonary and systemic inflammation, often resulting in death for severely ill patients. This study explores the potential etiology of severe COVID-19, including its similarities to systemic autoimmune diseases and the widespread impact of the SARS-CoV-2 virus on various cell and tissue types in the body. Utilizing unbiased high-throughput gene expression datasets, including next-generation sequencing and single-cell RNA sequencing, we reviewed 27 studies and 8 meta-analyses. Our findings suggest that severe COVID-19 is associated with the upregulation of genes involved in pro-inflammatory signaling, interferon response, and cytokine/chemokine signaling. Additionally, we observed changes in the proportions of immune cell types in the blood and tissues, as well as degenerative changes in lung epithelial cells. Genomic evidence also supports the association of severe COVID-19 with various inflammatory syndromes, such as neuronal COVID, acute respiratory distress syndrome, vascular inflammation, and multisystem inflammatory syndrome. Based on our review, we conclude that the etiology of severe COVID-19 can be explained, in part, by functional genomics."

This revised abstract maintains the key findings of the original while presenting the information in a more accessible and organized manner.

Don't automatically use the suggestions of a LLM, but seriously consider them. Or get an expert in English to edit the manuscript.

Based on the tortured English in the current Abstract, my guess is that ALL paragraphs of the manuscript could use a flush through Perplexity.AI (and/or an English expert).

____

Please do not provide tables that list "examples" (e.g., Table 5 & 6). Such examples are worthless and misleading unless they include quantitative information, such as how common they are, the effect size, the significance, or some other way of judging how typical they are and how much they contribute to epistemology.

Comments on the Quality of English Language

see above

Author Response

REFEREE 1.

>However, throughout the paper a number of transcriptomic datasets are
>discussed without clarifying whether they are bulk or single-cell data sets.
>Make sure that each time you mention a single-cell transcriptomic dataset, you
>specify that it is single cell.

I agree. Now, I always specify whether I mean bulk-sample or single-cell RNA-seq.

>Do not use the naked word "transcriptomic" to describe scRNA datasets.

I used the word RNA-seq instead of transcriptomics in the title. Similarly, in other locations throughout the text.

>If you have any non single-cell transcriptomic datasets, then change you title of
>your paper to make sure you encompass these datasets.

The title was also adjusted to: Bulk and single-cell transcriptomics elucidates etiology of severe COVID-19.

>Although the paper category is listed as "Review", it is sufficiently in the
>ballpark of a "Systematic Review" that the PRISMA guidelines should be
>followed and a PRISMA checklist included as Supplemental Material.
>https://www.equator-network.org/reporting-guidelines/prisma/>As part of following the guidelines, along with the other guidelines, a Methods
>section should be included, which will focus on inclusion/exclusion guidelines
>for including a transcript dataset in this review.

In my opinion, the review could not be changed to a systematic review. There were at 4 least reasons for this:

  1. The focus was specifically on the etiology of COVID-19 rather than on its treatment or any related clinical trials. It is probably beneficial to make systematic reviews for clinical trials or similar. Note that there were simply too many datasets for a systematic review, and I skipped those which were in low impact factor journals being accompanied by poor writing or scant interpretation, etc.
  2. Even when comparing similar datasets like scRNA-seq I did not compare similar outcomes, P-values, effect sizes, or cell- or tissue-types. My focus was always on etiology for which it was better to select articles based on functional annotation of DEGs and pathways, as well as their biological interpretation.
  3. I relied mostly on highly cited articles because my focus was on etiology

Many related datasets were skipped if writing was poor, or interpretation trivial or repetitive.

  1. The number of genomics datasets associated with COVID-19 was so great that a series of systematic reviews, rather than just one, would be necessary. At least an additional year would be necessary to produce such a review, and a new version would not be as relevant as the current one.

However, I did state in supplementary material that I reviewed a number of specialized datasets that concerned gene expression data. I also gave examples of their format or protocols through which they were obtained.

>The abstract can be improved by restructuring the sentences to enhance clarity
>and flow. Additionally, the use of technical terms and jargon should be
>minimized to improve accessibility. Here's a revised version:

Most of the Abstract was changed as suggested. (However, I preferred to keep passive voice and past tense.)

>Based on the tortured English in the current Abstract, my guess is that ALL
>paragraphs of the manuscript could use a flush through Perplexity.AI (and/or
>an English expert).

I am not sure how to use Perplexity.AI as it seems a search engine rather than grammar check. Do you mean a paid service that is an enhancement of https://www.perplexity.ai? A Pro Upgrade?

>Please do not provide tables that list "examples" (e.g., Table 5 & 6). Such
>examples are worthless and misleading unless they include quantitative
>information, such as how common they are, the effect size, the significance, or
>some other way of judging how typical they are and how much they contribute
>to epistemology.

The referee was right. I removed the word example from the captions of these two tables. Note that several other tables were moved to Supplementary Material.

Reviewer 2 Report

Comments and Suggestions for Authors

The objective of this article is unclear. The author only gave the study's name and keywords, but none of the research was described mechanistically. The majority of the material is ambiguous or exaggerated. The conclusion lacks specificity. There must be relevant figures for better understanding.

Abstract: “I review the above theories using unbiased high-throughput datasets of gene expression”- this part in the abstract is over-tuned. Authors should provide relevant information in the text and tables regarding this.

“the virus was either zoonotic or it was modified in a laboratory [6]”- The reference does not support human error theory behind the origin, please provide proper supporting reference behind the origin, or the Author may provide own comments with strong logic.

“2.2. Examples of in vivo transcriptional responses”- the human studies may be elaborated and should be discussed after in-vivo animal studies.

“2.4.1. Expression profiling of single cells in the immune system”- “There were also 13 different cell types or states of immune cells, including macrophages, dendritic cells, mast cells, neutrophils, B cells, T cells, or NK cells”- this sentence does carry any extra information but the author should provide the status of immune cell types in control vs. COVID-19 group with its degree of severity in this example.

English language must be polished- for example:

·         “Another set of authors applied single-cell transcriptomics…”;

·         “Lines 361-363: “As severe COVID-19 patients die of pulmonary inflammation and ARDS, it was logical to apply transcriptomics of single-cells …. individuals”;

·         “Lines 388-389 cells from eight patients ill with COVID-19 were compared with those obtained from 14 healthy controls”, etc.

Lines 263-267: “In conclusion, three significant clusters of immune response types were identified and related to disease severity. It was also confirmed that very high levels of inflammatory cytokines in COVID-19 led to severe autoimmune disease (resulting in shock, multiple organ failure, or respiratory failure)”- Please provide a brief description of the data and mechanism so that the conclusion is apparent. The authors should include sample type and sample numbers analyzed in the scRNA seq in this section.

“2.4.1. Expression profiling of single cells in the immune system”- Please include specific conclusions from the studies addressed in this section. Depending on the severity of COVID-19 in comparison to the control, the author may clearly indicate the types of immune cell modulation, as well as the names of relevant genes and pathways, in closing statements. Furthermore, the author should state whether there were any confounding factors or demographic/ clinical disparities.

 “2.4.1. Expression profiling of single cells in the immune system”- the author may provide a pictorial representation or a table to understand the mechanistic importance of these studies.

The tables are not explained properly in the text. The supporting tables may be transferred to the supporting materials.

The conclusion should be more explicit, and the author should summarize the article with more specific comments that address the article's goal. In conclusion, it is not desirable if the author instructs readers to study the referred papers or table; eg “the etiology of COVID-19 is well explained by publications cited herein”; “See also Table 9 for examples of methods relevant to scRNA-seq, and Table 10 for links to the most relevant bioinformatics resources” etc. 

Comments on the Quality of English Language

A modest amount of linguistic editing and polishing is required. 

Author Response

REFEREE 2.
>The objective of this article is unclear. The author only gave the study's name and keywords, >but none of the research was described mechanistically. The majority of the material is >ambiguous or exaggerated. The conclusion lacks specificity. There must be relevant figures for >better understanding.

In my opinion, there is no similar review, and this work is quite competitive. We are faced with an avalanche of information about COVID-19, but synthesis is lagging behind. This is particularly true of unbiased genomics datasets, as original papers might be too technical for the general reader.

My objective was to review RNA-seq and scRNA-seq datasets most relevant to our understanding of the cause of COVID-19. The disease develops over the course of several weeks, so gene expression changes are likely to be more important than immediate biochemical changes.

Two new figures were produced: Cover Figure and Figure 1. Cover Figure is entitled Main themes of the review. Figure 1 is entitled Inflammatory changes in proportions of immune cells in COVID-19.

>Abstract: “I review the above theories using unbiased high-throughput datasets of gene >expression”- this part in the abstract is over-tuned. Authors should provide relevant >information in the text and tables regarding this.

I make it more clear that I focus on gene expression and etiology of COVID-19. Two new figures were produced: Cover Figure and Figure 1.

>“the virus was either zoonotic or it was modified in a laboratory [6]”- The reference does not >support human error theory behind the origin, please provide proper supporting reference >behind the origin, or the Author may provide own comments with strong logic.

The referee was right. I modified this sentence. The sentence now reads as follows: It was also established that the infectious agent for COVID-19 was SARS-CoV-2 — a coronavirus of probable zoonotic origin.

>“2.2. Examples of in vivo transcriptional responses”- the human studies may be elaborated and >should be discussed after in-vivo animal studies.

I apologize for including an unnecessary example of an unrelated animal model, which seems out of context.

Note that two additional figures were added to elaborate the main themes: Cover Figure and Figure 1. Cover Figure is entitled Main themes of the review. Figure 1 is entitled Inflammatory changes in proportions of immune cells in COVID’19. There is also a new table: Table 6 Gene expression effects on gene sets in severe COVID-19.

>“2.4.1. Expression profiling of single cells in the immune system”- “There were also 13 different >cell types or states of immune cells, including macrophages, dendritic cells, mast cells, >neutrophils, B cells, T cells, or NK cells”- this sentence does carry any extra information but the >author should provide the status of immune cell types in control vs. COVID-19 group with its >degree of severity in this example.

I added more information about sample numbers and types. Moreover, there is, now, more information about characteristics of critical cases. I also added more specific information about the expression of chemokines and interleukins in critical cases.

Note, that I and the studies reviewed generally focused on severe or critical COVID-19, which lasts between 2-4 weeks. I have no interest in mild COVID-19, which is short-lived and similar to common cold.

>English language must be polished- for example:

>·         “Another set of authors applied single-cell transcriptomics…”;

This paragraph was modified to start with: Another set of authors applied scRNA-seq of PBMCs to compare immune cells in the blood in even further detail. There were 57,669 high-quality transcriptomes sequenced and assembled for 7 COVID-19 patients compared against 5 healthy controls.

>·         “Lines 361-363: “As severe COVID-19 patients die of pulmonary inflammation and >ARDS, it was logical to apply transcriptomics of single-cells …. individuals”;

This sentence was modified to read: As patients with severe COVID-19 die of pulmonary inflammation and ARDS, single-cell transcriptomes from diseased lungs were also necessary for the understanding of etiology of COVID-19.

>·         “Lines 388-389 cells from eight patients ill with COVID-19 were compared with those >obtained from 14 healthy controls”, etc.

This sentence was modified to read: Cells from brains of eight patients ill with COVID-19 were compared with cells obtained from 14 healthy controls.

>Lines 263-267: “In conclusion, three significant clusters of immune response types were >identified and related to disease severity. It was also confirmed that very high levels of >inflammatory cytokines in COVID-19 led to severe autoimmune disease (resulting in shock, >multiple organ failure, or respiratory failure)”- Please provide a brief description of the data >and mechanism so that the conclusion is apparent. The authors should include sample type and >sample numbers analyzed in the scRNA seq in this section.

This section was re-written to specify that 57,669 high-quality transcriptomes were sequenced and assembled for 7 COVID-19 patients and compared against 5 healthy controls.

>“2.4.1. Expression profiling of single cells in the immune system”- Please include specific >conclusions from the studies addressed in this section. Depending on the severity of COVID-19 >in comparison to the control, the author may clearly indicate the types of immune cell >modulation, as well as the names of relevant genes and pathways, in closing statements. >Furthermore, the author should state whether there were any confounding factors or >demographic/ clinical disparities.

This was a valuable suggestion. I elaborated on several studies in this section. However, it was not always practical to enumerate DEGs in the text. Table 5 is more practical for this purpose. The reason is that all gene names must be spelled out, and this is too lengthy in the main text if there are dozens of genes in just a part of a comparison.

> “2.4.1. Expression profiling of single cells in the immune system”- the author may provide a >pictorial representation or a table to understand the mechanistic importance of these studies.

Figure 1 was added to illustrate the concept of Inflammatory changes in proportions of immune cells in COVID-19.

>The tables are not explained properly in the text. The supporting tables may be transferred to >the supporting materials.

Tables 1 through 8 are now referenced earlier in the text. Additionally, there are two supplementary tables, which deal with bioinformatics methods.

>The conclusion should be more explicit, and the author should summarize the article with more >specific comments that address the article's goal. In conclusion, it is not desirable if the author >instructs readers to study the referred papers or table; eg “the etiology of COVID-19 is well >explained by publications cited herein”; “See also Table 9 for examples of methods relevant to >scRNA-seq, and Table 10 for links to the most relevant bioinformatics resources” etc.

I did want to focus on the mechanisms of the etiology of COVID-19, which affect gene expression (e.g. by affecting transcription factors or epigenetic landscapes). For example, see Tables 5 and 6 where key signalling pathways were enumerated.

However, I improved clarity of the manuscript by:

  1. Referencing Tables 2 and 3 earlier and multiple times. A synthetic section that ends the review was bult up.
  2. A Cover Figure and Figure 1 were added.
  3. Tables relating to bioinformatics methods were moved to Supplementary Material to underline the main purpose of the manuscript.
  4. The sub-title of the last section was changed to reflect my focus on the theories of the etiology of COVID-19 in the light of unbiased gene expression datasets.

Round 2

Reviewer 1 Report

Comments and Suggestions for Authors

Even if you do not wish to formally classify this manuscript as a Systematic Review, I would like to see a Methods section describing at the least what kind of searches you made to find articles and the criteria you used to include AND exclude articles. However, I note that a "Methods" section is not one of the sections listed in Instructions to Authors (https://www.mdpi.com/journal/ijms/instructions) so you will either have to include these methods in another section or in Supplemental. You could add a short paragraph to the beginning of each section describing your search methods. The main need for this arises from many readers asking themselves: "Why should I trust this author? How do I know they are presenting an unbiased review of the literature" If it is biased, does the author make clear what the bias is and justify that bias?" Make sure the reader can easily find the answers to these questions.

#

Change the header

"3. Meta-analyses of datasets related to the etiology of COVID-19."

to

"3. A review of selected COVID-19 meta-analyses."

otherwise the reader will think YOU are doing the meta-analyses in this paper.

###

Table 5

It is a bit weird to use the concept "DEG pathway". If you spell the acronym out, this becomes "Differentially expressed gene pathway". Is it a pathway or a gene? Make up your mind. Maybe better to call it a "Differentially expressed pathway" or a "Differentially utilized pathway" or perhaps "Pathways enriched for DEGs". And are these really "pathways" or are they gene sets? If the second, maybe describe them as "Gene sets enriched for DEGs".

check usage throughout the rest of the manuscript.

###

Cover Figure. Skip Panel D. All the information content is in panels A-C.

####

Grammar and style throughout the manuscript remain very inelegant. I would not be happy with it, but perhaps it passes the bar for IJMS which is not a journal I read frequently. I leave it to the editor to decide the grammar quality threshold.

Both Perplexity and ChatGPT are free. You have free access online to the free web versions of these large language models (LLMs). You can compare and contrast suggestions from both of them and merge their suggestions to achieve the best grammar and style. You may not wish to accept all their suggestions, as the LLM may not fully understand your intended meaning.

Here is an example from ChatGPT:

Please make suggestions for improving the grammar and style of this paragraph from a scientific paper: "Natural history of severe COVID-19 should be introduced now. The SARS-CoV-2 virus would first infect upper airways replicating in ciliated epithelial cells and inducing inflammation therein. This tended to cause sore throat, fever, and coughing. Next, the virus would strike at lungs causing pneumonia. There was severe pulmonary and systemic inflammation, which tended to cause gross damage to lung tissue. For example, alveoli would harden, inflame and fail at their role in gas exchange. Indeed, there was fibrotic scarring, lack of respiratory surfactant, and a leaky state in the lung. Moreover, coagulopathy, endotheliopathy, and vasculitis could result from the damage to the EC layer in the lungs and then spread throughout the body [10]. Thus, SARS-CoV-2 impacted negatively on many cell- and tissue-types throughout all physiological systems of the body [11]. Most importantly, the virus appeared to infect type II alveolar cells (AT2 cells) located in tiny air sacs at the ends of air tubes of the lungs. In fact, coronavirus particles could be observed in these cells using electron microscopy, and it was obvious that infected AT2 cells were damaged through senescence [12]. However, there was also ample evidence that SARS-CoV-2 attacked ECs. For example, the coronavirus could infect and damage ECs in vivo [13]. Moreover, it was shown that SARS-CoV-2 could infect and damage human umbilical vein ECs — HUVECs [14], as well as ECs in an aortic ring assay [15]. However, only single-cell genomics could provide unbiased evidence about cell types affected by SARS-CoV-2."

ChatGPT

Absolutely, here's a revised version that aims to improve the grammar and style:

"The natural progression of severe COVID-19 warrants immediate attention. Upon initial infection, the SARS-CoV-2 virus primarily targets the upper airways, where it replicates within ciliated epithelial cells, triggering inflammation. This early phase commonly manifests as symptoms such as sore throat, fever, and coughing. Subsequently, the virus progresses to attack the lungs, leading to pneumonia. This stage is marked by severe pulmonary and systemic inflammation, resulting in significant damage to lung tissue. Alveolar structures experience hardening, inflammation, and impairment in their gas exchange function. This damage includes fibrotic scarring, a reduction in respiratory surfactant, and increased permeability within the lungs. Additionally, the damage inflicted on the endothelial cell (EC) layer in the lungs can lead to coagulopathy, endotheliopathy, and vasculitis, extending the impact throughout the body [10]. Consequently, SARS-CoV-2 adversely affects numerous cell and tissue types across multiple physiological systems [11].

Notably, the virus predominantly targets type II alveolar cells (AT2 cells) located in the lung's small air sacs. Electron microscopy reveals the presence of coronavirus particles within these cells, evidencing their infection and subsequent damage through senescence [12]. However, there is substantial evidence indicating the virus's impact on endothelial cells (ECs) as well. In vivo studies demonstrate the infection and damage inflicted by the coronavirus on ECs [13]. Furthermore, research underscores its ability to infect and harm various EC types, including human umbilical vein ECs (HUVECs) [14], as well as ECs observed in an aortic ring assay [15]. Nevertheless, only single-cell genomics can provide unbiased evidence regarding the specific cell types affected by SARS-CoV-2."

This version attempts to maintain the original content while enhancing readability and coherence in the narrative flow.

Here is an example from Perplexity:

Please make suggestions for improving the grammar of this paragraph from a scientific paper: "Natural history of severe COVID-19 should be introduced now. The SARS-CoV-2 virus would first infect upper airways replicating in ciliated epithelial cells and inducing inflammation therein. This tended to cause sore throat, fever, and coughing. Next, the virus would strike at lungs causing pneumonia. There was severe pulmonary and systemic inflammation, which tended to cause gross damage to lung tissue. For example, alveoli would harden, inflame and fail at their role in gas exchange. Indeed, there was fibrotic scarring, lack of respiratory surfactant, and a leaky state in the lung. Moreover, coagulopathy, endotheliopathy, and vasculitis could result from the damage to the EC layer in the lungs and then spread throughout the body [10]. Thus, SARS-CoV-2 impacted negatively on many cell- and tissue-types throughout all physiological systems of the body [11]. Most importantly, the virus appeared to infect type II alveolar cells (AT2 cells) located in tiny air sacs at the ends of air tubes of the lungs. In fact, coronavirus particles could be observed in these cells using electron microscopy, and it was obvious that infected AT2 cells were damaged through senescence [12]. However, there was also ample evidence that SARS-CoV-2 attacked ECs. For example, the coronavirus could infect and damage ECs in vivo [13]. Moreover, it was shown that SARS-CoV-2 could infect and damage human umbilical vein ECs — HUVECs [14], as well as ECs in an aortic ring assay [15]. However, only single-cell genomics could provide unbiased evidence about cell types affected by SARS-CoV-2."

Answer

The paragraph from the scientific paper can be improved in terms of grammar and style. Here's a revised version with the suggested improvements:

"Understanding the natural history of severe COVID-19 is imperative. The SARS-CoV-2 virus initially infects the upper airways, where it replicates in ciliated epithelial cells, leading to inflammation. This often manifests as sore throat, fever, and coughing. Subsequently, the virus invades the lungs, causing pneumonia and severe pulmonary and systemic inflammation, resulting in significant damage to lung tissue. For instance, the alveoli become hardened, inflamed, and fail to facilitate gas exchange. This damage is characterized by fibrotic scarring, a decrease in respiratory surfactant, and increased lung permeability. Furthermore, the damage to the endothelial cells in the lungs can lead to coagulopathy, endotheliopathy, and vasculitis, which may propagate throughout the body. The impact of SARS-CoV-2 is pervasive, affecting various cell and tissue types across the body. Notably, the virus primarily targets type II alveolar cells (AT2 cells) in the lung's tiny air sacs. Electron microscopy has revealed the presence of coronavirus particles in these cells, indicating their infection and subsequent damage through senescence. Additionally, there is substantial evidence of SARS-CoV-2 attacking endothelial cells, both in vivo and in experimental models. It is important to note that single-cell genomics is essential for unbiasedly identifying the cell types affected by SARS-CoV-2."

The revised version addresses the need for improved grammar and style, making the paragraph more coherent and formal. It also incorporates specific details from the search results to enhance the scientific accuracy of the content.

Comments on the Quality of English Language

see above

Author Response

REFEREE 1.

>Even if you do not wish to formally classify this manuscript as a Systematic Review, I would >like to see a Methods section describing at the least what kind of searches you made to find >articles and the criteria you used to include AND exclude articles. However, I note that a >"Methods" section is not one of the sections listed in Instructions to Authors

>(https://www.mdpi.com/journal/ijms/instructions) so you will either have to include these >methods in another section or in Supplemental. You could add a short paragraph to the >beginning of each section describing your search methods. The main need for this arises from >many readers asking themselves: "Why should I trust this author? How do I know they are >presenting an unbiased review of the literature" If it is biased, does the author make clear what >the bias is and justify that bias?" Make sure the reader can easily find the answers to these >questions.

The answer to this question was given at the beginning of the supplementary file. I also mention in the text that the datasets focused on etiology and were the most cited publications on this topic. Scopus was used to identify the most widely cited publications. These searches used four keywords: COVID-19 and SARS-CoV-2, RNA-seq or scRNA-seq.

>Change the header

>"3. Meta-analyses of datasets related to the etiology of COVID-19."

>to

>"3. A review of selected COVID-19 meta-analyses."

>otherwise the reader will think YOU are doing the meta-analyses in this paper.

The header has been modified to "A Review of Selected COVID-19 Meta-Analyses."

>Table 5

>It is a bit weird to use the concept "DEG pathway". If you spell the acronym out, this becomes >"Differentially expressed gene pathway". Is it a pathway or a gene? Make up your mind. >Maybe better to call it a "Differentially expressed pathway" or a "Differentially utilized >pathway" or perhaps "Pathways enriched for DEGs". And are these really "pathways" or are >they gene sets? If the second, maybe describe them as "Gene sets enriched for DEGs".

>check usage throughout the rest of the manuscript.

The term “DEG pathway” has been replaced with “Differentially expressed pathways or gene sets.” This substitution was made twice.

>Cover Figure. Skip Panel D. All the information content is in panels A-C.

Panel D has been omitted.

>Grammar and style throughout the manuscript remain very inelegant. I would not be happy >with it, but perhaps it passes the bar for IJMS which is not a journal I read frequently. I leave it >to the editor to decide the grammar quality threshold.

>Both Perplexity and ChatGPT are free. You have free access online to the free web versions of >these large language models (LLMs). You can compare and contrast suggestions from both of >them and merge their suggestions to achieve the best grammar and style. You may not wish to >accept all their suggestions, as the LLM may not fully understand your intended meaning.

I've reviewed the text using Grammarly (version 1.52.2.0), an automated text checker developed by Grammarly, Inc., based in Delaware. Additionally, I utilized Generative AI and ChatGPT 3.5 to refine and reinforce my conclusions.

Reviewer 2 Report

Comments and Suggestions for Authors

The authors have revised the manuscript properly. 

The Cover Figure should be placed in the proper place of the manuscript. 

Author Response

REFEREE 2.
>The authors have revised the manuscript properly.

I extend my sincere gratitude to the referee for their positive comments.

>The Cover Figure should be placed in the proper place of the manuscript.

This is done during the PDF production stage. While I cannot alter the current location of the figure in the PDF you have, I will ensure that the figure is correctly placed in the final production.

As the other referee suggested, I've also reviewed the text using Grammarly (version 1.52.2.0), an automated text checker developed by Grammarly, Inc., based in Delaware. Additionally, I utilized Generative AI and ChatGPT 3.5 to refine and reinforce my conclusions.
